

# Revisiting the DeepWater Horizon spill: High resolution model simulations of effects of oil droplet size distribution and river fronts

Lars R Hole[1], Knut-Frode Dagestad[1], Johannes Röhrs[1], Cecilie Wettre[1], Vassiliki H. Kourafalou[2], Ioannis Androulidakis[2], Matthieu Le Hénaff[2], Heesook Kang[2], and Oscar Garcia-Pineda[3]

[1]Norwegian Meteorological Institute, Allegt. 70, 5007 Bergen, Norway
[2]University of Miami, Rosenstiel School of Marine and Atmospheric Science, Miami, FL, USA
[3]WaterMapping, Gulf Breeze, FL, USA
**Correspondence:** Lars Robert Hole (lrh@met.no)

**Abstract.** An open source ocean trajectory framework, OpenDrift, is used to simulate the 2010 DeepWater Horizon oil spill. Metocean forcing data are taken from the GoM-HYCOM 1/50° ocean model with realistic river input and ECMWF global forecasts of wind and wave parameters with 1/8° resolution. OpenDrift includes the integrated oil drift module OpenOil, which includes a number of relevant processes, such as emulsification, wave entrainment, and droplet formation. This takes account of

the actual oil type/properties, using the ADIOS oil weathering database of NOAA. We investigate the effect of using a newly developed parameterization for oil droplet size distribution, compared to a traditional algorithm. Although the algorithms provide different distributions for a single wave breaking event, we find that the net difference after long time simulations is negligible, indicating that the outcome is robust regarding the choice of parameterization. In both cases, the size of the droplets controls how much oil is present at the surface and hence are subject to wind and Stokes drift. The oil droplet sizes are also

relevant for the biological impact. Next, the effect of removing river outflow in the ocean model is investigated in order to showcase effects of river induced fronts on oil spreading. A consistent effect on the amount and location of stranded oil is found, and considerable impact is seen on the location of the surface oil patch.

# 1 Introduction

The presence of both shelf and open sea dynamics makes the Northern Gulf of Mexico (NGoM) a topographically and dynam-
ically complex and interesting study area, in the presence of intense oil exploration. Interactions of the Mississippi River (MR) plume and the Loop Current (LC) system were found important on the transport and fate of oil during the 2010 DeepWater Horizon (DWH) incident (Kourafalou and Androulidakis, 2013; Le Hénaff et al., 2012).

The NGoM, and especially the western and central region around the MR Delta, is a major source of oil and natural gas in
the U.S. According to the U.S. Energy Information Administration, more than 45% of total U.S. petroleum refining capacity and 51% of total natural gas processing plant capacity is located along the Gulf coast (EIA). Oil leaks and accidents, such as the





explosion on the DWH platform in 2010 (at 28.737ºN, 88.366ºW), have released significant quantities of hydrocarbons (Crone and Tolstoy (2010); McNutt et al. (2011)) in the sensitive marine environment around the MR Delta, and over the LouisianA TEXas (LATEX) and Mississippi Alabama FLoridA (MAFLA) shelves (Kourafalou and Androulidakis, 2013) (Fig. 1).

Androulidakis et al. (2018b) carried out a field experiment near the Taylor Energy Site which is located in the vicinity of

the MR outflow region over the NGoM and near the DWH site (approximately at 28.938ºN, 88.978ºW, Fig. 1). This multi-platform observational experiment was conducted in April 2017 to investigate the main transport pathways from the Taylor Site and toward the NGoM continental shelves and offshore, toward the Gulf interior. Results indicated that the surface transport was determined by the MR plume extension over the Taylor Energy Site and the river induced fronts in combination with local circulation, prevailing winds and broader regional dynamics (LC system). The drifters deployed during the field experiment

in tandem with satellite data, drone imagery, wind measurements, and marine radar derived currents and images efficiently described three major transport pathways, in agreement with the three major circulation patterns of the MR plume (Schiller and Kourafalou, 2010; Androulidakis et al., 2018a; Schiller et al., 2011; Schiller and Kourafalou, 2014).

The drifters followed the two prevailing coastal currents associated with MR plume dynamics (downstream/upstream moving westward/eastward of the Mississippi Delta) and an offshore pathway under the influence of basin-wide circulation. Near

the Taylor site, the existence of multiple river fronts influence the fate of oiled waters, preventing the transport of hydrocarbon toward the delta like a natural boom barrier, trapping and directing the oil either westward or eastward in agreement with Kourafalou and Androulidakis (2013), who showed a similar interaction during the DWH accident. *In situ* thermohaline measurements around the Taylor Energy Site and across the river front showed that the MR plume near the Taylor Site was 5m to 10m deep, while the clearer ocean water column was characterized by a 40 m upper-ocean homogeneous layer, mainly

controlled by temperature.

In this study, an open source Lagrangian oil drift model, OpenOil, has been used to simulate the DWH oil spill evolution. OpenOil takes into account all important factors that influence the short term drift of an oil slick such as metocean forcing (including Stokes drift), emulsification, evaporation and vertical entrainment and mixing. Simulations are initiated from satellite observations and point sources. The effect of using two different oil droplet size distribution on the horizontal drift and vertical

mixing is discussed. Next, the study showcases how NGoM oil pathways are influenced by river plume circulation and river induced fronts. We also investigate whether the use of realistic daily river discharge has a significant effect on the simulated location of the Surface Oil Patch (SOP) and stranding of oil.

## 2   Methods and data

### 2.1   Shapefiles of surface oil patch

As part of the efforts made by the National Oceanic and Atmospheric Administration (NOAA) to asses the extent and impact of the DWH spill, participants on this team analyzed hundreds of satellite images (microwave and optical) and produced oil extent delineations throughout the lifetime of the spill. Classifications derived from the satellite analysis of the DWH SOP can be accessed through the NOAA-ERMA website (ERM). The satellite analysis of the DWH extent include not only the



oil slick boundary delineation but also the classification of thin or thick areas of the oil layer within the slick (also called 'actionable' oil). We used the oil thickness classifications derived from the satellite analysis for our modeling study. Oil particles were seeded uniformly within the region enveloping the thick and thin oil slicks with no distinction. This oil extent and thickness classification dataset is available on the NOAA-ERMA website in the form of GIS shapefiles containing polygons.

The shapefiles were used for both initialization of the oil drift simulations and for verification of results.

## 2.2  Metocean forcing

In the cases presented here, the ocean circulation fields come from a high-resolution (1/50º,  1.8 km) configuration of the Hybrid Coordinate Ocean Model (HYCOM) in the Gulf of Mexico (GoM), using daily river forcing and data assimilation (GoM-HYCOM 1/50º). The HYCOM model has a flexible, hybrid vertical coordinate system, in which the distribution of

vertical layers is optimized by using: isopycnal layers in stratified water columns, sigma terrain-following layers in regions with sharp topography, and isobaric in the mixed layer and very shallow areas Bleck (2002). More information about the HYCOM model is available in the model user's manual (https://hycom.org) and the references therein. The GoM-HYCOM $1/50°$ covers the entire GoM and uses 32 vertical levels. The model configuration is similar to the one used by Le Hénaff and Kourafalou (2016), with the realistic river forcing parameterization developed by Schiller and Kourafalou (2010). The

river discharge data were obtained from the Army Corps of Engineers and the U.S. Geological Survey (USG). It is nested at the open boundaries in the operational Global HYCOM (GLB-HYCOM) simulation run at the Naval Research Laboratory at the Stennis Space Center (GLB-HYCOM expt_90.8, (HYC). The atmospheric forcing is based on the 3-hourly winds, thermal forcing and precipitation fields from the European Centre for Medium-Range Weather Forecasts (ECM), with spatial resolution of 0.125º (see below). The model assimilates satellite observations of Sea Surface Temperature and Sea Surface Height, and

in situ observations of temperature and salinity from buoys, cruises, surface drifters, Argo floats and XBT casts. More details about the model configuration can be found in Le Hénaff and Kourafalou (2016) and Androulidakis et al. (2018a). For the present study, we performed two simulations: one with the attributes mentioned above, called *Reference* simulation, and one called *No river*, in which the salinity fronts have been removed by shutting off the river discharge, setting precipitation to zero, and turning off the assimilation of salinity profiles.

The ECMWF provides global daily forecasts at 0 and 12 UTC with 0.125º resolution. Recent model upgrades have improved the overall performance of the forecasting system throughout the medium range. Further details on model description and verification can be found e.g. in Ehard et al. (2016), Haiden et al. (2016) and at ECM. Here, ECMWF daily forecasts were used as atmospheric upper boundary conditions for the GoM-HYCOM 1/50 as well as for providing air temperature and wind drag for the OpenOil simulations with a 3 hourly time step.

The third generation spectral wave model, WAM is well known, see Group (1988) and Haiden et al. (2016). The WAM model computes two-dimensional wave spectra, with 25 frequencies and 24 directions. From the two-dimensional spectra, several parameters are computed, including significant wave height, peak wave period, mean wave period, peak wave direction and mean wave direction. The wave parameters are computed for total sea, and for wind sea and swell separately (Haiden et al., 2016). The operational daily WAM forecasts used here are forced by the ECMWF atmospheric forecasts. WAM model





output with 0.125° horizontal resolution are downloaded from ECMWF and used here for estimating horizontal Stokes drift and vertical mixing of the oil with a 3 hourly time step.

We focus here on two experimental periods 20-27 May and 2-10 July. According to the models applied here, during the first period, wind speed varied between 0.1 and 7.2ms$^{-1}$ and the significant wave height varied between 0.1 and 1.2m. In the second

period, the corresponding values were 5 to 12 ms$^{-1}$ and 0.1 to 3.2m.

## 2.3 The oil drift model OpenOil

OpenOil is used to model the transport and fate of the oil released from DWH. OpenOil is based on the OpenDrift trajectory modeling framework (Dagestad et al., 2018), developed at the Norwegian Meteorological Institute and available as open source software from Ope. It is used as operational oil spill contingency and search and rescue model in Norway. OpenOil has been

evaluated against drifter and oil slick observations in the North Sea (Jones et al., 2016; Röhrs et al., 2018). Details of the particle tracking model are given in Dagestad et al. (2018), and model physics that are specific to oil transport and fate are documented in Röhrs et al. (2018).

OpenOil is an integrated oil drift model consisting of sub-models for specific physical processes like wave entrainment of oil (Li et al., 2017c), vertical mixing due to oceanic turbulence (Visser, 1997), resurfacing of oil due to buoyancy (Tkalich and

Chan, 2002), and emulsification taking account for oil properties (Lehr et al., 2002). The resurfacing is a function of oil density and droplet size following Stokes Law, and thereby the model physics are very sensitive to the specification of the oil's droplet size.

Several algorithms are implemented to describe the oil's droplet size spectra, based on previous published parameterizations. The first option is based on the work of Delvigne and Sweeney (1988) (DS88), manifesting a power-law droplet size number

distribution as a function of droplet size, with an exponent of -2.3, i.e. there are many more small droplets than large droplets. Transferring this to a volume size distribution, as needed for practical oil spill simulation that follows the mass of the oil spill, the exponent becomes 0.7, i.e. there is more volume in the few large droplets than in the many small droplets. The typical droplet sizes range from $1\,\mu m$ to $1\,mm$.

A second option to describe the droplet size distribution is based on Li et al. (2017b) (Li17), which takes the oil viscosity and

the oil-water interfacial tension into account. This parameterization describes a log-normal law for the number size distribution, and the resulting volume size distribution exhibits a peak at an intermediate droplet size of about $100\,\mu m$, depending on oil type and environmental conditions. Similar types of droplet size spectra have been developed and observed, confirming that there is a maximum in oil volume at a particular droplet size (Johansen et al., 2015; Li et al., 2017a).

With regard to horizontal drift, three processes are considered: Any element, whether submerged or at the surface, drifts

along with the ocean current. Elements are further subject to Stokes drift corresponding to their actual depth. Surface Stokes drift is normally obtained from a wave model, and its decline with depth is calculated as described in Breivik et al. (2016). Oil elements at the ocean surface are additionally moved with a factor of 2% of the wind. Together with the Stokes drift (typically 1.5% of the wind at the surface), this sums up to the commonly found empirical value of 3.5% of the wind speed (Schwartzberg, 1971). The physical mechanism behind this wind drift factor is not obvious, and is discussed in Jones et al. (2016).





The three horizontal drift components may lead to a very strong gradient of drift magnitude and direction in the upper few meters of the ocean. For this reason, it is also of critical importance to have a good description of the vertical oil transport processes, which are the sum of the following processes:

Oil elements at the surface, regarded as being in the state of an oil slick, may be entrained into the ocean by breaking waves. The entrainment of oil droplets depends on both the wind and wave (breaking) conditions, but also on the oil properties, such as viscosity, density and oil-water inter facial tension. The buoyancy of droplets is calculated according to empirical relationships and the Stokes law following Tkalich and Chan (2002), dependent on ocean stratification based on temperature and salinity from the ocean model, and the viscosities and densities of oil and water.

In addition to the wave induced entrainment, the oil elements are also subject to vertical turbulence throughout the water column, described using a random-walk scheme based on the turbulent eddy diffusivity from the ocean model. In addition to the vertical and horizontal drift, weathering of the oil also has to be considered. OpenOil interfaces with the open source ADIOS oil library, developed by NOAA (https://github.com/NOAA-ORR-ERD/OilLibrary, Lehr et al. (2002)). In addition to state-of-the art parametrization of weathering processes such as evaporation, emulsification and dispersion, this software contains a database of measured properties of almost 1000 oil types from around the world. As oils from different sources or wells have vastly different properties, such a database is of vital importance for accurate results. The ADIOS oil library is also used by the NOAA oil drift model PyGNOME (https://github.com/NOAA-ORR-ERD/PyGnome).

OpenOil take into consideration weathering processes that are dominating in the initial oil spill period of 2-3 days. Long-term weathering processes such as dissolution, sedimentation and microbial degradation are not considered in this study.

## 3  Results

We focus here on the DWH spill of April-July 2010. A first set of simulations are carried out to investigate the effect of oil droplet size distribution. A second set focuses on the effect of river induced fronts. According to Crone and Tolstoy (2010), the average flow rate from the oil well between 22 April and 3 June 2010 was estimated to 0.1 m$^3$ sec$^{-1}$. After the riser was removed and until the leak was sealed on 15 July, the flow rate increased to 0.12 m$^3$ sec$^{-1}$, corresponding to 10,368 m$^3$ day$^{-1}$. Although our purpose here is not to provide an accurate quantitative estimate of the oil fate, but rather to discuss the processes, we make an assumption of how much oil was present at the sea surface at the initialization of the simulation. The lifetime of oil at the sea surface depends heavily on oil properties as well as environmental conditions such as temperature of ocean and air, wind and waves, as described above. According to our mass balance calculations for the DWH spill, using the Light Louisiana Sweet oil type from the NOAA oil library and environmental conditions as described above, it seems reasonable to assume that 80% of the oil mass is removed from the surface after 10 days. This is within the range in our simulations that is typically 60 to 95% (see examples of mass balance plots further down). The simulations for May 2010 are initialized by seeding 48730 super-particles in a polygon obtained from NOAA shapefiles. Each particle represents initially 1 m$^3$ oil. A continuous point source at the sea floor seeds an additional 8460 particles (8460 m$^{-3}$) per day during the simulation. After June 3rd these numbers are increased by 20% to 10368 m$^3$ day$^{-1}$.



**Table 1.** Percentages of stranded oil particles for the May 20-27 simulations.

|  | *Reference* simulation | *No River* simulation |
|---|---|---|
| West of MR Delta | 6.9 | 0.5 |
| MR Delta area | 4.0 | 3.4 |
| East of MR Delta | 1.2 | 2.1 |
| Total | 12.1 | 6.0 |

Around 20-25 May 2010 there was a significant outflow of the Mississippi River (Fig. 2) and part of the SOP was entrained along the LC resulting in a formation popularly referred to as the "tiger tail" (Fig. 3). The OpenOil simulation shown on top in Fig. 3 is carried out using the classical DS88 oil droplet size distribution (Delvigne and Sweeney, 1988). Fig. 3 lower panel shows the results from repeating this simulation using the new Li17 formulation. In Fig. 7 we show the mass balance during

seven days for the Li17 simulation. There is virtually no difference between DS88 and Li17 and only the former is shown here. Both formulations result in about the same fraction of oil at the surface (about 50%) after 7 days, with moderate wind speeds of up to 7.2ms$^{-3}$. We also see that patches of thick oil (where the super-particles retain nearly 100% of their mass) are visible over a larger area (Fig. 3). Larger oil droplets will rise faster to the surface, and DS88 provides a much higher fraction of large droplet after one hour compared to Li17 (Fig. 4, upper panels). However, it turns out that the DS88 and Li17

provide very similar volume distributions after a 24 hr test simulation (Fig. 4, lower panels). Due to this small difference, the DS88 simulation results in just marginally more particles stranded after seven days (13.2 vs 12.8 %), particularly west of the Mississippi Delta (Figs. 3). A higher fraction of oil at the surface provides more efficient transport by wind and waves towards the shore and larger likelihood of stranding. For both simulations, the oil particles quickly loose 40-50% of their mass, mostly through evaporation.

Fig. 5 shows the geographical distribution of particle diameters. It is apparent that smaller particles are present outside the MAFLA shelf, probably because the oil particles have been subject to more wind action in the last 12 hours (Fig. 6).

We chose the Li17 formulation for two types of simulations: *Reference* (all forcing data) and No river (all forcing data except for river runoff and precipitation). Two periods were chosen for these experiments: a high river discharge period with variable winds (20-27 May) and a relatively lower discharge period with persistent westward winds (2-10 July). The purpose

is to investigate the effect of the salinity fronts by using the ocean circulation from the *No river* simulation, in which the precipitation and river discharge are turned off, while atmospheric and wave forcings are kept the same. In Figs. 8, 9, 10 and 11 we see how the inclusion of river discharge in the ocean forcing can have opposite (and sometimes counter-intuitive) effects on oil transport. Table 1 summarizes the difference of the *Reference* and *No River* simulations for May 20-27, whilst Table 2 shows corresponding values for the 2-10 July simulations.

Kourafalou and Androulidakis (2013), based on high-resolution ocean simulations over the NGoM, showed that the MR discharge peak around 20 May led to the formation of downstream (westward) and upstream (northeastward) plume areas that acted as a conduit for guiding oil toward the LATEX shelf and away from the MAFLA shelf, respectively (Fig. 9). In the 20-27





**Table 2.** Percentages of stranded oil particles for the July 2-10 simulations.

|  | *Reference* simulation | *No River* simulation |
|---|---|---|
| West of MR Delta | 10.6 | 7.8 |
| MR Delta area | 16.7 | 27.2 |
| East of MR Delta | 20.8 | 20.1 |
| Total | 48.1 | 55.1 |

May period the river plume currents created a strong "bulge" that tended to turn waters clockwise around the Delta, with some waters moving westward. In addition, the offshore GoM circulation (Loop Current and eddies) removed the riverine waters offshore, toward the GoM interior, forming the so-called "tiger tail" pathway. The removal of the MR input in the *No River* experiment during this period (Fig. 8) allowed the spreading of oil toward the western MAFLA shelf due to the absence of this anticyclonic bulge; the stranded oil along the MAFLA coasts is more apparent in the *No River* case in comparison to the *Reference* experiment. In contrast, the amount of stranded oil is less along the western coasts (90W-91W) in the *No River* experiment due to the absence of the downstream MR plume pathway. Moreover, the "tiger tail" signature is weaker in the *No River* simulation. Kourafalou and Androulidakis (2013) showed that the strength of this MR offshore jet could have been an important factor in forming the "tiger tail" oil distribution pattern as also confirmed from satellite and drifter data (Walker et al., 2011).

The second simulation period (2-10 July) was right after another high discharge period (although not as high as in May), promoting again a buoyancy-driven downstream current. This tendency was supported by downwelling-favorable winds (Kourafalou and Androulidakis, 2013), resulting in a clear westward transport of both low-salinity and oil containing waters, along a narrow band (of similar width) close to the LATEX coast and surrounding the Mississippi Delta; extensive coastal areas of stranded oil are apparent along the western coasts in the *Reference* experiment (Figs. 10 and 11). The removal of MR input (*No River* experiment) led to weaker downstream currents both close to the Delta (89.5W) and along the western coasts (west of 90W) and thus less stranded oil over the same region. The anticyclonic bulge, common in strong discharges and source of the downstream current (Kourafalou et al., 1996), is completely absent in the *No River* experiment. As a result, more stranded oil was computed closer to the Delta, inside Louisiana Bight in the *No River* case. It seems like the absence of the anticyclonic bulge that was able to lead surface oiled waters directly west of the Louisiana Bight allowed the accumulation of oil very close to the Delta. In contrast, smaller differences between the two experiments are detected over the MAFLA region due to the weaker upstream currents during early July (Kourafalou and Androulidakis, 2013). Due to slightly higher wind speed during this period, we can notice that most particles have lost 70-90% of their mass due to evaporation shortly after release (Fig. 10).



## 4   Discussion and conclusions

We have carried out several simulations of the DWH oil spill with high resolution forcing data and a Lagrangian oil spill model. Simulations were initialized from satellite observations of the SOP, and a continuous point source with a realistic spill rate at the sea floor.

Our results indicate that the use of a realistic oil droplet distribution will have significant effect on both vertical and horizontal distribution of the oil, when wind speeds are typically 5-7 ms$^{-1}$ and breaking waves can be expected Fig. 3. Both formulations of the oil droplet size distribution result in the characteristic "tiger tail" shape of the SOP for the period 20-27 May 2010, and significant stranding in the delta west of the Mississippi River mouth in line with the observed SOP (see also http://archive.nytimes.com/www.nytimes.com/interactive/2010/05/01/us/20100501-oil-spill-tracker.html) and
(Kourafalou and Androulidakis, 2013).

Both droplet size formulations that are used here (DS88 and Li17) result in similar size distributions after some time of simulation, as seen in Fig. 4. Li17 prescribes a maximum droplet diameter in the volume spectra as seen in Fig. 4, which is due to two regimes in the size spectrum where small droplets are limited by viscous effects, and larger droplets by oil-water interfacial tension (Li et al., 2017a). This causes a peak in the volume spectra as seen in laboratory experiments with repeated
mixing and wave breaking from the surface. The DS88 spectra does not prescribe such a maximum, using a power-law that increases towards larger droplets in the volume spectra. However, the time-integrated simulations in OpenOil still produce a maximum in the droplet size spectra. The reason for this is the repeated wave breaking at the surface, which is more pertinent to large droplets that quickly rise to the surface. Hence, the description of buoyancy driven resurfacing and wave breaking in the oil spill model, together with the DS88 droplet size spectrum for individual wave breaking events, produces similar results to
a more advanced droplet size spectra that explicitly prescribes a maximum in the volume size spectra. Note that both methods result in approximately the same maximum droplet size after 24 hours (Fig. 4).

A realistic description of droplet formation is required to describe the effects of an oil spill on the environment. Fig. 5 shows that the oil spill transport during the DWH spill favors a transport of small particles towards the northeast, while larger oil droplets follow the paths towards southwest and southeast. As a results of their low buoyancy and turbulent mixing, smaller
particles are mixed into deeper parts of the ocean and subject to ocean currents at depth (Röhrs et al., 2018). Larger particles experience stronger buoyancy and are subject to surface currents or return to the surface slick. As wind and waves only affect the near-surface drift, the part of the oil slick that forms large droplets is quickly separated from the small droplets which retain at larger depths. This will also impact the effect of the spill on the ecosystem: The parts of the oil spill at the surface is more hazardous to birds and the beach communities, while the small, submerged parts will have a substantially larger surface to
interact with water and plankton (Carroll et al., 2018).

Next, we evaluated the effect of realistic river discharge on the simulations. One might expect that removing the river discharge would always bring the oil nearer to the shore, but interactions are complex. The *No River* simulation for May 20-27 showed more stranding oil, in particular close to the Louisiana Bight, but less stranding oil further downstream, along the LATEX shelf. The removal of the MR input reduced the downstream currents that were responsible for the westward transport





of oiled waters along the LATEX shelf. The MR plume and the accompanying river fronts were responsible to either entrap oil close to the coasts (e.g. LATEX shelf) or keep oiled waters offshore (e.g. MAFLA shelf) due to the the formation of upstream currents (9). These results are in line with (Kourafalou and Androulidakis, 2013) and the NOAA SOP observations used here and shown in 8. It is also obvious that, in the *Reference* simulation, the oil particles are guided by the river fronts and they are

carried further away from the coast, pushed into the LC south of 28ºN and E of 88.5ºW (9).

The second simulation period (2-10 July) was right after a second high discharge period, promoting again a buoyancy-driven downstream current. This tendency is supported by downwelling-favorable winds, resulting in a clear westward transport of both low-salinity and oil-containing waters, along a narrow band close to the LATEX coast and surrounding the Mississippi Delta; extensive coastal areas of stranded oil are apparent along the western coasts in the *Reference* experiment. The removal

of MR input (*No river* experiment) led to weaker downstream currents both close to the delta (89.5W) and along the western coasts (west of 90W) and thus less stranded oil over the same region.

The simulations presented here were initiated by seeding oil particles evenly in a polygon defined by NOAA satellite products, in addition to a continuous point source at the sea floor. The next possible step is to initiate the simulations from satellite products which contain information about oil film thickness in addition to area, and hence also quantify the amount of oil at

the surface.

To the best of our knowledge, this is the first time the importance of the effect of river fronts on oil slick transport in the gulf has been demonstrated using high resolution models.

*Code availability.* https://github.com/OpenDrift/opendrift

*Data availability.*

*Code and data availability.*

*Sample availability.*

*Video supplement.*



*Author contributions.*

*Competing interests.*

5 *Disclaimer.*

*Acknowledgements.* This research was made possible by a grant from The Gulf of Mexico Research Initiative (award "Influence of river induced fronts on hydrocarbon transport," GOMA 23160700). Atmospheric and wave data were kindly provided by the European Center for Medium-Range Weather Forecasts (ECMWF).



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



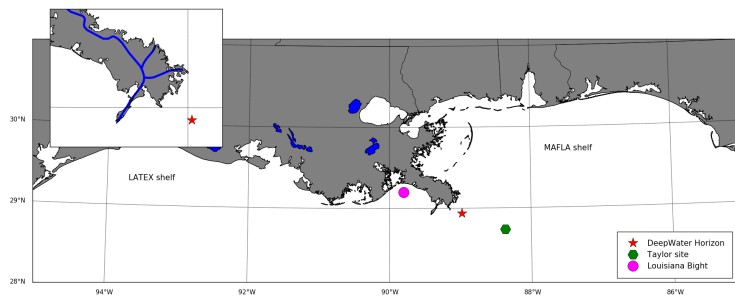

**Figure 1.** Map of the Northern Gulf of Mexico showing the geographical locations mentioned in the text. The map insert shows the Mississippi River (MR) delta in blue with the three major river passes that release MR water into the Gulf.

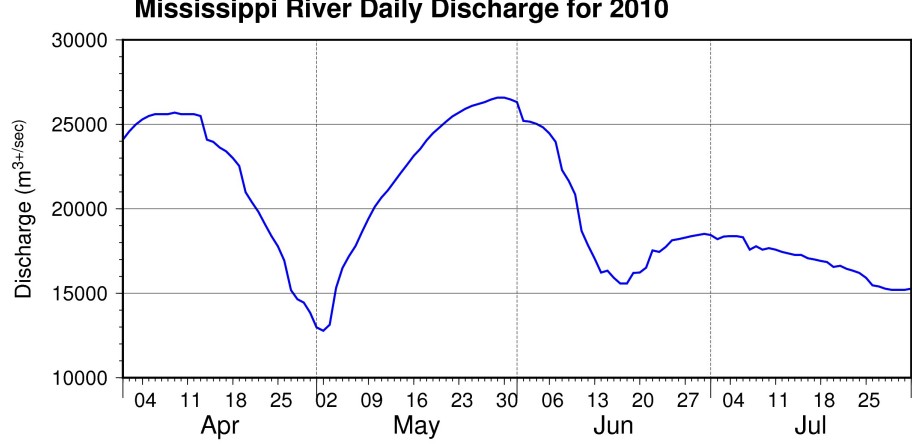

**Figure 2.** Discharge from Mississippi River in the Northern Gulf of Mexico during late spring and early summer 2010. Data kindly provided by U.S. Geological Survey (USGS) and U.S. Army Corps of Engineers





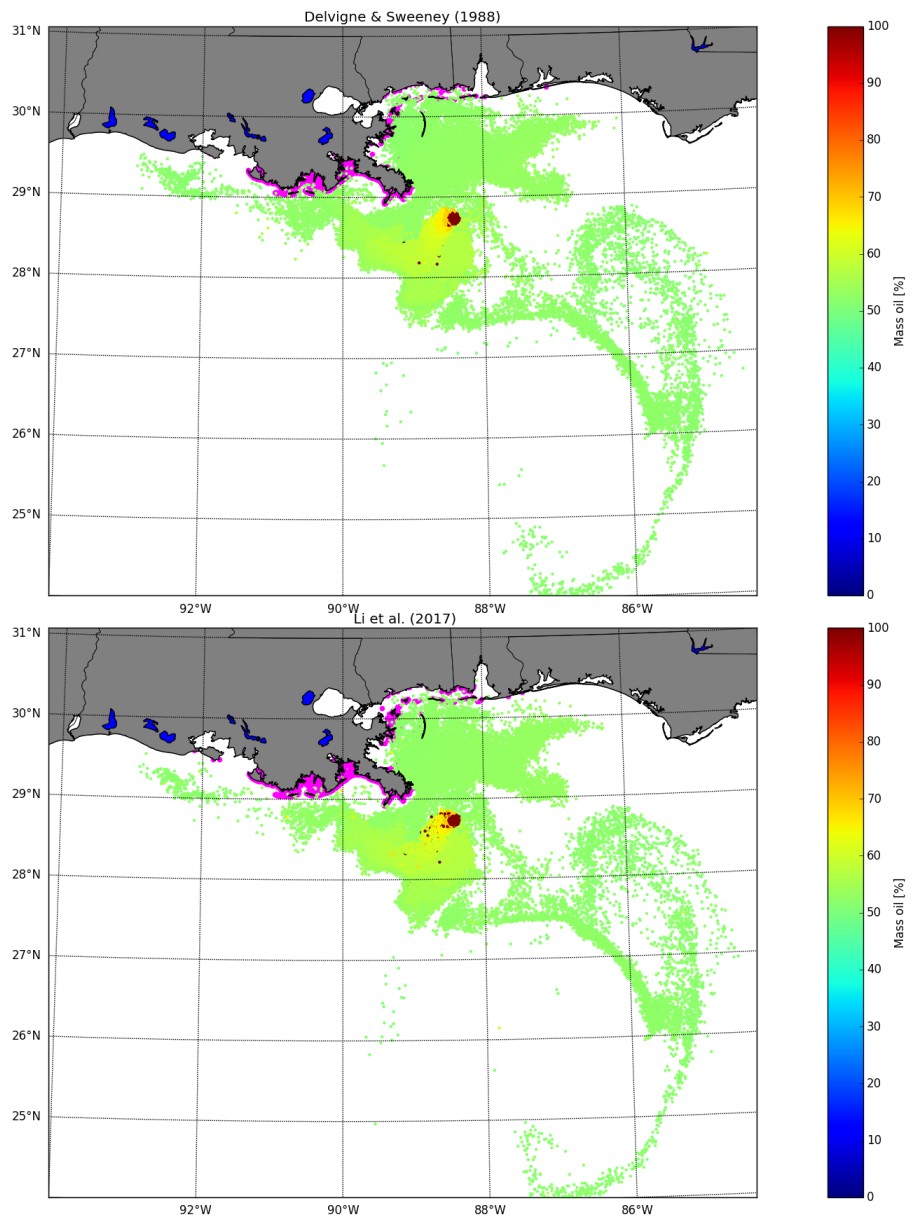

**Figure 3.** OpenOil simulation for 20-27 May 2010, using the Delvigne and Sweeney (1988) oil droplet size distribution (upper - 13.2% stranded oil), and the Li et al. (2017c) distribution (lower - 12.8% stranded oil). Only surface oil particles are shown. Patch colors are the fraction of mass left in the particles. Magenta areas indicate stranded oil.





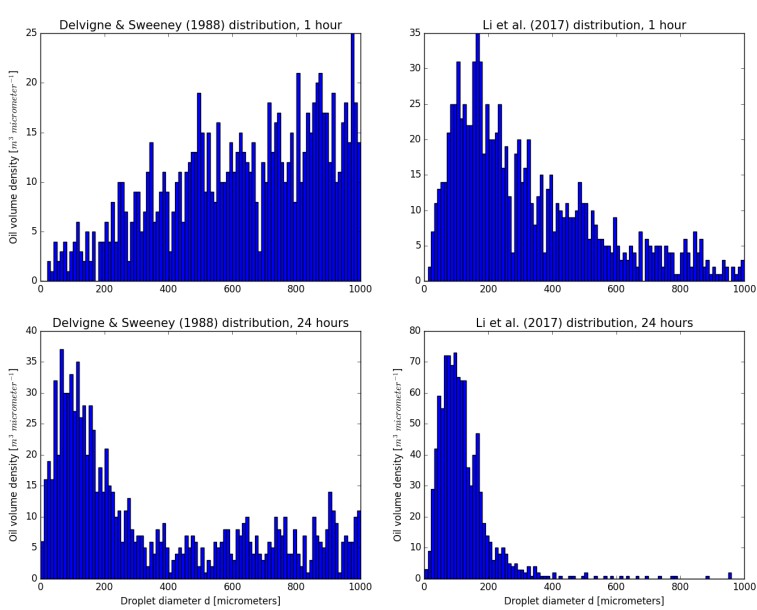

**Figure 4.** Oil droplet volume histogram for 1000 particles after 1 hours using the *Light Louisianna Sweet, BP* oil type in OpenDrift during
8 ms$^{-1}$ wind. Delvigne and Sweeney (1988) formulation at top left and Li et al. (2017c) at top right. Bottom panels show corresponding
distributions after 24 hours.





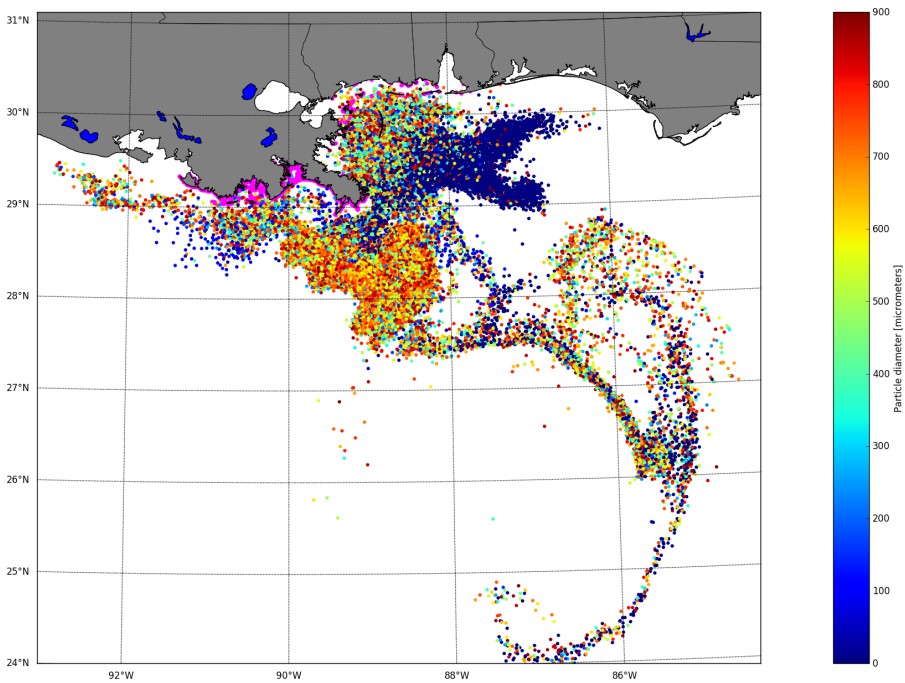

**Figure 5.** End condition of the *Reference* simulation 20-27 May 2010, showing all active particles (at surface and submerged). Same simulation as in Fig. 3 lower panel. Stranded oil is shown in magenta. The color scale shows diameter of the oil droplets.





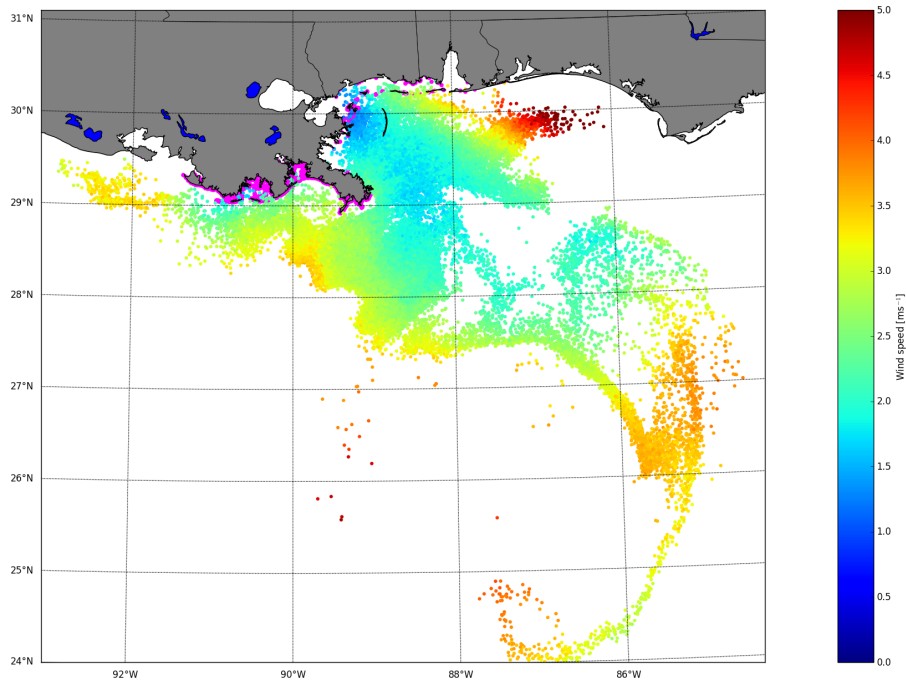

**Figure 6.** Same as Fig. 5, but the color scale shows the average wind speed experienced by the particle during the last 12 hours.

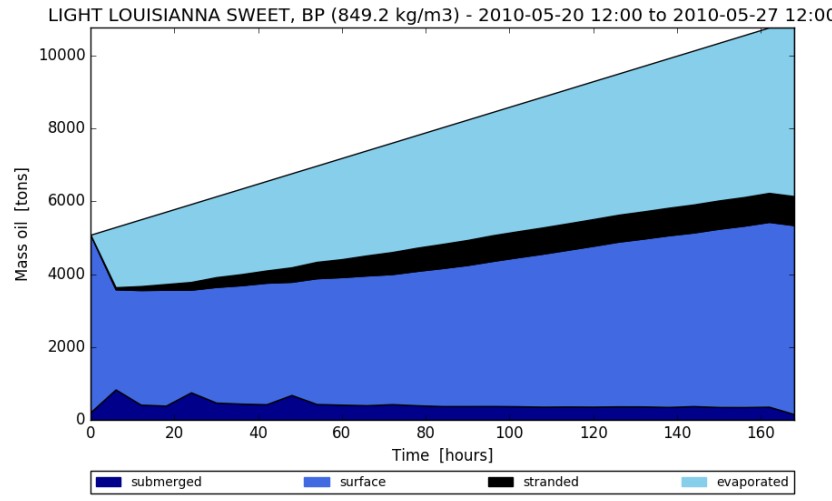

**Figure 7.** Mass balance of oil in the OpenDrift simulation shown in Fig. 2. The Li et al. (2017c) oil droplet size distribution is used.





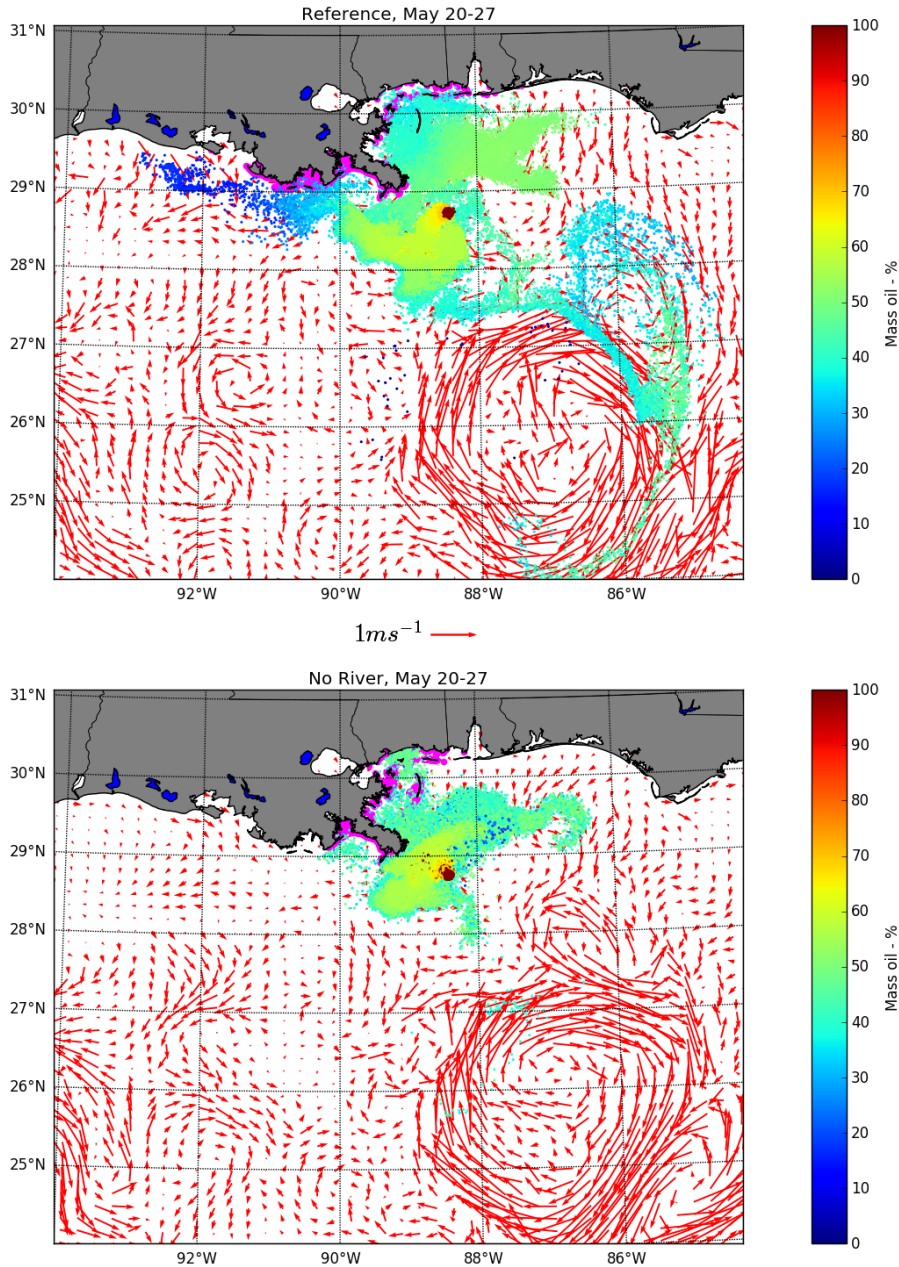

**Figure 8.** End condition of the simulation 20-27 May 2010, showing active (at surface) and stranded oil particles. Stranded oil is shown in magenta. The color scale indicate how much mass is left in each particle. Red arrows are the GoM-HYCOM 1/50 forcing surface currents at the last time step of the simulation (every 5th data point shown). *Reference* simulation at top (12.1% stranded oil), and *no river* simulation below (5.1 % stranded oil). See also Table 1.





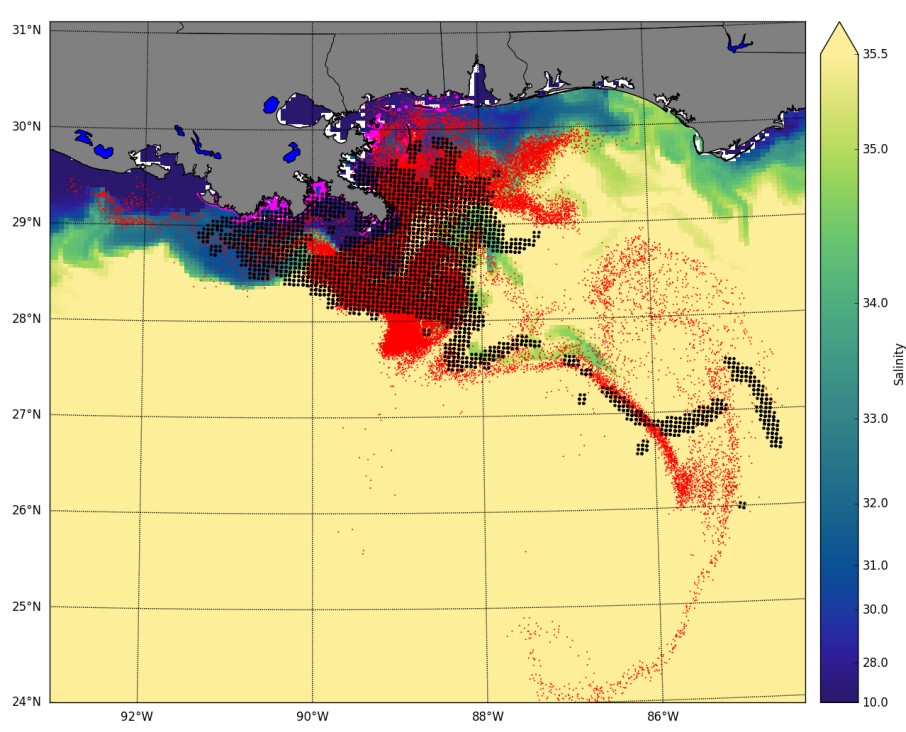

**Figure 9.** End condition of the *Reference* simulation 20-27 May 2010 (Same simulation as in Fig. 8, upper panel), showing active particles at surface as red dots dots and the corresponding observed surface oil patch (NOAA shape file) as black dots. Modeled stranded oil is shown in magenta. The color scale shows sea surface salinity in the forcing data.




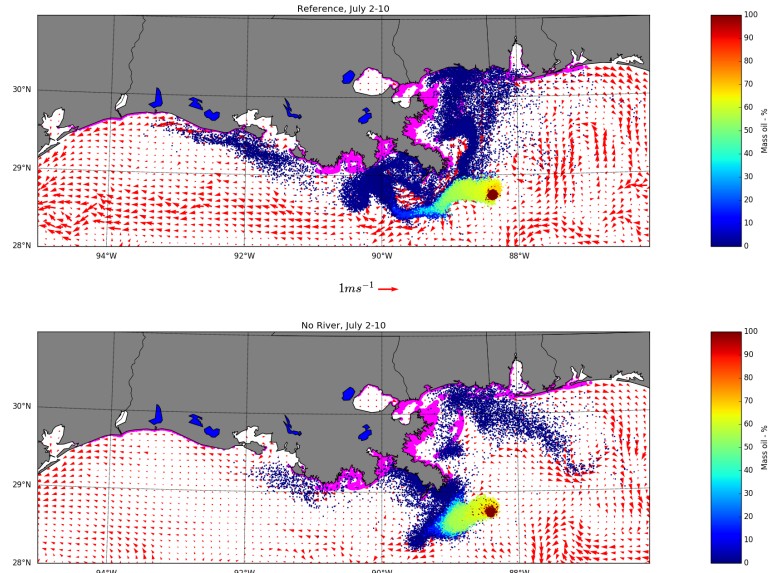

**Figure 10.** End condition of the simulation 2-10 July 2010, showing active (at surface) and stranded oil particles. Stranded oil is shown in magenta. The color scale indicate how much mass is left in each particle. Red arrows are the GoM-HYCOM 1/50 surface currents at the last time step of the simulation (every 2nd data point shown). *Reference* simulation at top (48.1% stranded oil), and *no river* simulation below (55.1% stranded oil). See also Table 2.

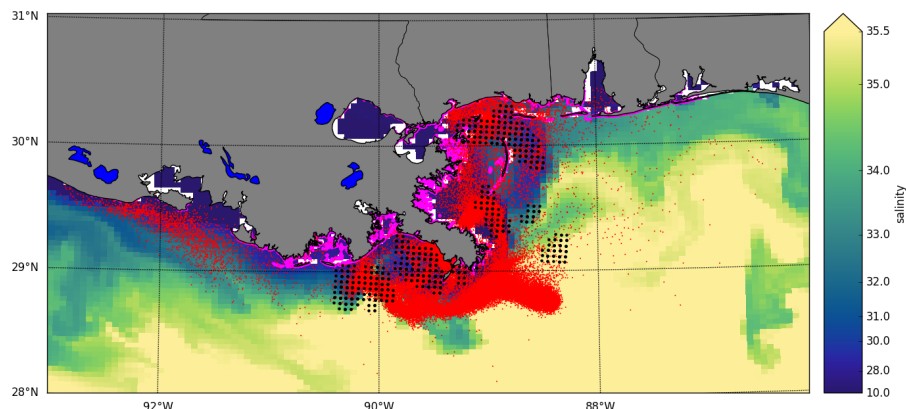

**Figure 11.** End condition of the *Reference* simulation 2-10 July 2010 (Same simulation as in Fig. 10, upper panel), showing active particles at surface as red dots dots and the corresponding observed surface oil patch (NOAA shape file) as black dots. Stranded oil is shown in magenta. The color scale shows sea surface salinity in the forcing data.