# Peer review of "Revisiting the DeepWater Horizon spill: High resolution model simulations of effects of oil droplet size distribution and river fronts"

_Ocean Science, 2018_

## Referee Comment (RC1) · Anonymous Referee #1 · 24 Dec 2018

Hole and the coworkers present a simulation study of surface oil spreading during the DeepWater Horizon oil spill. The authors use an open source oil trajectory model to test the surface spreading with two different oil droplet size distributions, and the cases with/without the Mississippi River fresh water plume. With their modeling results, the authors conclude that the surface oil spreading and the final size distribution are almost independent of the selection of the initial oil size distributions as a long-term process. The model shows the impact of MR plume on the surface transport of oil slicks.

This study is generally useful; however, it lacks the description of the model initiation on droplet sizes and the discussion of the oil transport in the water column. Particularly,

the authors did not mention how different oil size distributions were set in the model and what are driving forces that were used to calculate the oil sizes, which I believe is the key to any transport processes in the oil spill modeling. This is a No. 1 weakness of the paper because the authors are trying to understand the impact of initial oil sizes on the model results. One important question is that whether the oil droplets were released from the oil wellhead or were just released at the water surface (although the authors mentioned this in the end of the paper). The author mentioned in the end of Section 2.1 that shapefiles were used for initialization of the oil drift simulation, which makes me wonder if the oil were released at the surface, not at the wellhead. In Page 2, line 24, the authors claimed that the simulation was initialized from satellite observations and point sources. So, it is not clear how oils with different size distributions were put in the simulation and how they were calculated. If two different oil size distributions were used at the wellhead, it is hard to believe the surface oil slick would have no (or negligible) difference according to the authors' conclusion.

Another weakness of the paper is that the discussion is descriptive, not quantitative. Apparently the usefulness of the qualitative discussion on helping oil spill response is limited. The authors compare the spreading area of the oil slick on the surface, but it is not clear the masses of oils were transported in the spatial domain. Also, in the discussion of the oil droplet sizes in Fig. 5, it is mentioned these are all particles at the surface and submerged. Can you quantify the percentage of the oil at the surface and in the water column? Both sizes, numbers, and volume are important parameters, which should be discussed. A 3D plot might be useful. See a previous work:

North et al. (2011) Simulating Oil Droplet Dispersal From the Deepwater Horizon Spill With a Lagrangian Approach.

About the oil droplet sizes: the authors claim that the oil droplet sizes are similar after 1 hour (Fig. 4). First, it is not clear what does this 1 hour mean, from the initial condition? What is the initial condition (oils were released at the surface)? If the oil were released at the wellhead, how long it would take for the oil surfacing? How is the surface time

related to oil droplet sizes? Second, the two distributions in the lower panel are very different from the upper panel in my opinion. The lower-left panel has much more larger oil droplets in the system. If you integrate all of these larger droplets, the total volume of the oil is significant, at least on the same order of magnitude as in the smaller spectrum of the oil size distribution. Hence, I don't agree that the two distributions give the similar results. Again, the author must clarify whether the oils were released from wellhead or the surface. If at the surface, what is the total volume? What driving forces are used to calculate the size distribution? If the oils were released at the wellhead, because small oil droplets have much longer lifetime in the water column (they rise slower) than large ones, loss of the oil mass will be so different with different initial wellhead size distributions. Hence, the resulting sizes of surfacing oils at the water surface will need to be calculated with a fate and transport model. Do you include these processes in the model?

The paper also has many vague statements and topographic errors (see below). Some minor comments are also included here.

Page 1, line 9: in Abstract, "The oil droplet sizes are also relevant for the biological impact." Page 2, line 1: "significant quantities" Page 2, line 22, "all important factors", I would use "major factors" Page 2, line 26, "significant effect" Page 3, line 11, "Bleck (2002)" -> (Bleck, 2002) Page 3, line 15, "USG" -> "USGS" Page 3, line 17, missing ")" Page 4, line 16, "model physics", be specific. Page 5, line 14, "state-of-the-art" Page 6, line 4, "Fig. 7"? Page 6, line 5, "former"? Page 6, line 7, ms-3 -> ms-1 Page 6, line 7, define "super particles" Page 6, line 10, "very similar volume distribution". I do not agree. Page 6, line 11, "more particles", particle numbers or volume? Page 6, line 15, it will be interesting to check how different sizes of oils move with wind, especially those submerged in the water. Page 8, line 3, "realistic spill rate", be quantitative. Page 8, line 11-14. Not sure what the authors wanted to deliver here. Discussion of the oil droplet sizes is not adequate. The discussion is very qualitative. Page 8, line 30, "large surface" -> large surface area? Page 9, line 3-5, a few hyperlinks seem not

working. Page 9, line 10-11, be accurate describing coordinate. I would delete the last paragraph.

Towards the end of the paper, the author finally mentioned the oil particles were evenly distributed at the surface, and at the point source at the sea floor. Do you release same oil sizes at both surface and seafloor? How do you calculate these oil size distributions? Why do you just release oil particles at the seafloor? Without the information of how oil sizes were calculated and what oil fate and transport model was used, I think this manuscript is lack of fundamental information to justify the rational of the simulation. Hence, I do not recommend publication of this manuscript.

---

## Short Comment (SC1) · 7 Jan 2019

We thank the reviewer for very thorough comments to our manuscript. We do agree that the initialization of droplet size distribution needs to be described better and we are now preparing a better description. In the manus we describe that we release oil particles in a polygon at the surface (from satellite obs) + a point source at the sea floor. The release rates are quantified on p5, but we will move this further up. The size distribution is initially uniform and will develop over time, but we show that the DS88 and Li17 formulations give near identical size distributions after 24hours. We believe that this is a new and interesting result. We also refer to Röhrs et al. for detailed

description of the size distribution formulations.

The amount of oil at the surface and in the water column is shown in Fig 7.

One main conclusion from our work is the the two oil droplet size distribution formulations give near identical results. This is also a new result, and we will try to state this even clearer.

Sincerely, Lars R. Hole

―――――――――――――――――――――

---

## Referee Comment (RC2) · Anonymous Referee #2 · 20 Jan 2019

The goal of the article was to simulate the movement of the surface oil of the DWH on the water surface paying attention to the size distribution of droplets and the Mississippi River system. The article has many good contributions such as the comparison between various models for droplet formation and entrainment (Delvigne and Sweeney and Li et al 2017b,c). The OpenOil model seems promising. However, the writing of the article is poor and misleading, and I am not sure anyone could benefit from it, as the major findings are not supported by the needed explanation (for example what are floating droplets and what are submerged droplets?).

The article is too wordy. For example, the Introduction seems to focus on discussing

a couple papers related to the MR rather than providing a background for investigating the DWH spill in relation to existing works that modeled the DWH surface spill, such as the works of McFaddyen et al. (EOS, 2011) and Boufadel et al. (ES&T, 2014). In fact, only by comparing to such works that the authors can highlight their contributions. How do the results based on the more accurate HYCOM compare to results obtained using NCOM, NGOM, etc? The oil behavior appears to be an aftermath of the exercise, which is to highlight the role of the MR. One is led to wonder, what would be the fate of the oil if one assumes that the droplets are neutrally buoyant? What was the depth of mixing? In essence, there were too many things that the authors considered giving the impression that the article was put together hastily without a goal in mind. Detailed comments are below. "Next, the study showcases how NGoM oil pathways are influenced by river plume circulation and river induced fronts. We also investigate whether" It is preferable to stick to one tone of writing, either passive (the study discusses) or active (we investigate). Mixing the two is not common.

"As part of the efforts made by the National Oceanic and Atmospheric Administration (NOAA) to asses the extent and impact of the DWH spill, participants on this team analyzed hundreds of satellite images (microwave and optical) and produced oil extent delineations throughout the lifetime of the spill. Classifications derived from the satellite analysis of the DWH SOP can be accessed through the NOAA-ERMA website (ERM)."

Which team? The NOAA team or the authors? Also, why not writing that "we analyzed satellite images (microwave and optical) of the spill from the NOAA site ERMA" ? Why using such a wordy approach?

"The shapefiles were used for both initialization of the oil drift simulations and for verification of results."

By whom? By NOAA or by the current authors?

"For the present study, we performed two simulations: one with the attributes mentioned above, called Reference simulation, and one called No river, in which the salinity fronts have been removed by shutting off the river discharge, setting precipitation to zero, and turning off the assimilation of salinity profiles."

Please mention here in a sentence why the two choices were made. Assuming that one wants to evaluate the impact of the MR, how realistic would be to turn off the flow of the Mississippi? Wouldn't using the minimum flow be more realistic? Is the goal to conduct an academic exercise to evaluate the impact of the MR on the near shore hydrodynamics? If yes, then, this should be conducted only in a Monte Carlo framework as done by NOAA's Barker (EOS, 2011). Otherwise, the results and conclusions would be dependent on the hydrodynamic and climatological conditions during the DWH spill (i.e., not general).

"The physical mechanism behind this wind drift factor is not obvious, and is discussed in Jones et al. (2016)."

What was the summary from Jones et al. (2016)? The increase in oil drift is well understood for anyone who worked on transport. Based on simple momentum considerations, one could learn that anything that is on the top of the water surface (and has a smaller density) would move faster than the water beneath it if the wind is blowing (i.e., the whole idea of a sail boat !). Also, the fact that the oil viscosity is much higher than that of the water beneath it, makes the oil behaves locally as an "object". But if the authors have no idea why the oil moves faster than the water beneath it, I am not sure they should be conducting oil spill simulations using complicated models for evaporation, emulsification, dispersion, etc. Also, these processes were mentioned early on, and then were not discussed afterwards.

"According to our mass balance calculations for the DWH spill, using the Light Louisiana Sweet oil type from the NOAA oil library and environmental conditions as described above, it seems reasonable to assume that 80% of the oil mass is removed from the surface after 10 days. This is within the range in our simulations 30 that is typically 60 to 95% (see examples of mass balance plots further down). The simulations

for May 2010 are initialized by seeding 48730 super-particles in a polygon obtained from NOAA shapefiles. Each particle represents initially 1 m3 oil. A continuous point source at the sea floor seeds an additional 8460 particles (8460 m-3) per day during the simulation. After June 3rd these numbers are increased by 20% to 10368 m3 day-1."

The work by Boufadel et al. (2014, EST) specifically addressed this issue. See, for example, their Figure 2. They found that the oil disappears of the surface at around 20% per day. " The OpenOil simulation shown on top in Fig. 3 is carried out using the classical DS88 oil droplet size distribution (Delvigne and Sweeney, 1988)."

What was the thickness of the "Surface Layer" in the simulation? 1.0 m? 10.0 m? Obviously, this affects the meaning of concentration.

"Fig. 3 lower panel shows the results from repeating this simulation using the new Li17 formulation. In Fig. 7 we show the mass balance during seven days for 5 the Li17 simulation. There is virtually no difference between DS88 and Li17 and only the former is shown here."

Why discussing Figure SEVEN prior to Figures 4, 5, and 6? Should Figure 7 be made as Figure 4?

Table 2 provides oil deposited on the shorelines, and the amount needs to be compared with the work of Boufadel et al. (2014, ES&T).

How are the authors deciding what is on the surface and what is submerged? What was the cutoff depth? I find it unusual that they did not provide such information considering that GOMRI works including their own showed the importance of depth (on the order of centimeters) on the hydrodynamics.

"A realistic description of droplet formation is required to describe the effects of an oil spill on the environment"

True, but there is no citation for this statement.

"The parts of the oil spill at the surface is more hazardous to birds and the beach communities, while the small, submerged parts will have a substantially larger surface to 30 interact with water and plankton (Carroll et al., 2018)."

The authors cite a paper on the impact to the Northeast fisheries in spite of the HUGE amount of papers dealing with the DWH spill. It is pretty strange. Maybe they can start with the review by Short (2017, Archives of Environmental Contamination and Toxicology). Maybe they can peruse the GOMRI website?

"To the best of our knowledge, this is the first time the importance of the effect of river fronts on oil slick transport in the gulf has been demonstrated using high resolution models."

The work of the authors themselves addressed the impact of the MR on coastal hydrodynamics. Kourafalou, V. H. and Androulidakis, Y. S.: Influence of Mississippi River induced circulation on the Deepwater Horizon oil spill transport, Journal of Geophysical Research: Oceans, 118, 3823–3842, 2013. Le Hénaff, M. and Kourafalou, V. H.: Mississippi waters reaching South Florida reefs under no flood conditions: synthesis of observing and modeling system findings, Ocean Dynamics, 66, 435–459, 2016.
* * *

---

## Referee Comment (RC3) · Anonymous Referee #3 · 29 Jan 2019

**1   General Comments**

This paper presents simulations of oil transport during the Deepwater Horizon (DWH) accident using a Lagrangian Particle Tracking model with oil fate processes (OpenOil) coupled to an ocean circulation, wave, and wind model (GoM-HYCOM, WAM, and ECMWF), run under different freshwater discharge scenarios through the Mississippi delta system.  While such a study is important and in general the modeling systems used are adequate for the task, the present manuscript lacks the information needed to critically assess the modeling results.

[Figure]

Very broadly, the article lacks the clarity needed to understand both what the authors have done and how their models work. In Section 2, where the modeling system is described, references are missing to external model products (e.g., ECMWF results) and the text is unclear whether standard model products are used or whether custom simulations were conducted for this work. Where custom simulations are done, there needs to be better referencing to the base model and citations to any work validating the custom model. These problems should be easily resolved by re-writing this section to become much more clear. Indeed, poor writing quality is a problem throughout this manuscript, and most of the detailed comments below stem from vague or difficult to understand sentences throughout the text.

The greatest technical deficiency stems from the description of OpenOil and its usage. The detailed comments below give several, specific issues related to the handling of this model in the text. In summary, the following elements need to be improved before the model results can be critically evaluated:

- The authors need to better distinguish surface and sub-surface sources of oil and describe the chemical properties of the oil as initialized in the model. I assume the authors use the dead-oil representation of LSC from ADIOS for both surface and subsurface sources. This is ok, but they need to clearly state this and explain what the flow rate at the subsurface source means (it neglects all gas).

- The authors need to provide the equations for DS88 and Li17 since they are single-line, empirical formulas. This will greatly improve understanding for the reader.

- The authors need to explain how OpenOil uses DS88 and Li17 to predict a time-evolving droplet size distribution. Both of these equations are empirical formulas for the steady-state size distribution of submerged oil under various surface wave conditions. It is not obvious how these models produce time-variable droplet size distributions.

- Only Li17 predicts oil droplet size for a jet (e.g., the subsurface source), and technically, this is a different equation (different fit parameters) to the surface size distribution in that paper. What equation was used for subsurface oil? How was this done in the DS88 simulations?

- The fate processes considered in the model are applicable to surface oil. They can also be used for subsurface oil when dead oil is introduced into the model (which I believe the authors have done). However, dissolution, which is neglected in the present study, was a significant fate process during DWH and occurred very quickly. The authors are only justified to ignore it because they track only the low-solubility components of the oil.

After these important issues are resolved, the results can be critically evaluated. Throughout the results, the authors should carefully connect droplet sizes to mechanisms. For example, in point 25.) below the authors state that small droplets were subject to greater wind forcing. In what way? Perhaps this is true, but the mechanism is unstated. My reading of this does not make sense because I would assume smaller particles move away from the sea surface, where they are less impacted by winds. But, another reading could be that smaller particles are generated by stronger winds. Yet, the manuscript does not really explain it either way.

Overall, my conclusion is that this is an important study conducted by researchers who are clearly capable of handling the ocean circulation under different freshwater inflows. They should spend more time polishing the text and much more carefully explain the droplet size modeling, handle the complex thermodynamics of the oil and gas, and more carefully assess and describe the important fate mechanisms. The remainder of this review addresses specific elements of the manuscript text.

**2   Detailed Comments**

1. In the abstract, the authors describe the droplet size analysis and the key result (that the results are robust to the choice of algorithm). However, for the effects of river inflow, the abstract lacks the key result. It would be best to also summarize the main outcome of that work within the abstract.

2. The introduction is a collection of quite detailed sentences that do not flow together well. While each sentence is well referenced, I wonder whether the normal reader can follow these statements. For instance, for the Taylor Energy study, the paper reads, "The drifters deployed during the experiment...efficiently described three major transport pathways;" yet, the text does not introduce or describe the drifters before this key result statement. It would be better to say earlier in the paragraph that the Taylor Energy study included deployment of surface drifters and explain whether they were all released at the source? all at the same time? on what dates or in which season? what layer of water is tracked? As is, the reader really needs to know more about this study than is stated to understand the text. This is a specific comment that can be generalized to most of the statements in the introduction. This part of the paper could be significantly improved by more careful and tight writing.

3. Page 2, Lines 21-25. This section is a little misleading. In the abstract and later in the paper, the reader will realize that oil droplets are initialized at the surface and at the seabed. The model does include the key fate factors for surface oil, but does not include the key fate factor for the subsurface oil (dissolution). This section should be revised to remove this ambiguity. As is, it sounds like the model includes the key fate factors for all oil droplets released in the model; this is only true for the surface released oil or the list of factors stated in this sentence is incomplete (it should also include dissolution).

[Figure]

4. Page 2, Line 33. Include a citation with a link to the ERMA website and the date the data were accessed / downloaded.

5. Page 3, Line 2. "We used the oil thickness classifications derived from the satellite analysis for our modeling study." How were these used? The next sentence says that particles were seeded without respect to thickness. At least give the reader some hint here how thickness was used and where in the manuscript this concept will be expanded.

6. Page 3, Line 6. It seems that this sentence needs a citation to the source where the HYCOM model data, or at least numerical grid and initial conditions, were downloaded. The statements starting on Line 9 hint that the model simulations used in this paper may have been generated by the authors for this paper. Is that the case? If so, this should be explicitly stated. GoM-HYCOM is also an operational model, so a reader may assume that standard HYCOM products were used. The last sentence seems to make this more clear (maybe the authors use their own output). I recommend revising this paragraph to make it even more clear that the authors performed custom simulations using this modeling system, state what elements are from standard GoM-HYCOM products, clearly explain the initial conditions, and explain whether data assimilation is turned on for any of the present simulations.

7. Much of the text could be much better polished. A good example is the sentence starting on Page 3, Line 9, "The HYCOM model has..." In this sentence, the different layers are described. Use parallel grammatical structure: "Isopycnal layers in stratified water", "sigma terrain-following layers in sharp topography", then edit the next phrase to read "isobaric *layers* in the mixed layer and very shallow areas." Finally, the format of the citation needs to be within parentheses, e.g., (Bleck, 2002). While this does not impact the scientific merit of the paper, these types of careless mistakes distract from the impact of the paper and make

it more difficult for the reader to understand. As in point 2.) above, this is just one illustration of sloppy writing throughout the manuscript.

8.  Inconsistent usage of ECM and ECMWF (Page 3, Lines 18 and 25). Also, provide a citation for where and when the ECMWF data were downloaded.

9.  Page 3, Line 28. Oil simulations were performed with a 3-hourly time step; yet, the text says that ECMWF forcing data have a 12 hourly time step. How was this data sub-sampled? Sample and hold? Linear interpolation? Other?

10. Page 4, Line 1. Provide a citation to the data source for the WAM data that were downloaded from ECMWF.

11. Section 2.2. As one reads this entire section it is very difficult to separate modeling work done by the present authors from modeling work conducted elsewhere and used by the present authors. Perhaps it would be helpful for each model product to start with a statement that either reads something like, "Wave properties were downloaded from WAM model results available from ECMWF (citation)," or "We simulated wave forcing using the WAM model." Also, the acronym WAM is undefined.

12. Page 4, Line 18. "Droplet size spectra" should probably be "Droplet size distribution". Reserve the term "spectrum" to refer to energy distributions.

13. Page 4, Line 18ff. The Delvigne and Sweeney (1988) model must predict two parameters of the droplet size distribution: a characteristic droplet size and a variance parameter. The text appears to discuss the variance parameter, as given by a power-law relationship. I think this section would be much easier to understand if the equation is provided, e.g., $PDF = ad^b$. Then, the authors can also discuss the proportionality constant (i.e., $a$) and the state variable of the power law would be defined (presently, it is undefined, and I assume it is droplet size, $d$).

14. Page 4, Line 26. I am uncertain under what conditions Li17 would give $d_{50} = 100\ \mu$m. That is a very small size that would result from high mixing energy and low interfacial tension. Rather than state "depending on oil and environmental conditions"; instead state, "for an oil with $X$ interfacial tension and $Y$ viscosity, in a sea state of $Z$, ...". As is, this formulation is very misleading since it is difficult to get oil droplets this small without intervention (e.g., chemical dispersant application).

15. Page 4, Line 34. "physical mechanism...is not obvious." On the contrary, the physical mechanism for wind drift is well known and understood. Oil typically occupies a region very close to the water surface or on the surface (e.g., slicks). Numerical models of ocean circulation have large vertical layer thickness at the surface (about 50 cm or more) relative to the oil (about 1 mm to 10 cm). If the numerical model completely resolved the surface boundary layer, there would be no added wind drift. Because the circulation model integrates the top 50 cm to 1 m, there needs to be an empirical wind drift added. 3% of the wind is a typical value only because circulation models have very similar vertical layer thicknesses in their numerical discretization. If a model had a unique surface discretization, a different empirical factor would be needed.

16. Top of Page 5. Unusual usage of a colon at the end of the first paragraph. Replace with ""...the sum of oil entrainment by breaking waves and vertical turbulent diffusion." Also, "inter facial" should be "interfacial". Finally, the present grammatical construction makes it seem as if the ADIOS mechanisms are part of the processes describing vertical transport, which they are not. Instead, ADIOS is a 0-D model that predicts oil fate.

17. Page 5, Line 18. Dissolution is by no means a "long-term weathering process." Dissolution is largely complete in the time span of minutes to hours and much less than one day. Dissolution is ignored in surface fate models because it is

typically about 10 times less important than volatilization. For oil *not* in contact with the air/water interface (e.g., oil discharged at the seafloor), dissolution is the dominant fate process, and is the main fate mechanism for 100% of the methane discharged from the DWH and for about 25% of the total mass of petroleum spilled. This dissolution occurred as oil droplets and gas bubbles transported from the riser to the sea surface, a process that took about 4 to 12 hours duration (see papers by Ryerson in *GRL* and *PNAS*). Dissolution cannot be listed in this sentence as a long-term fate process, nor can it be completely ignored for DWH. The present authors may ignore dissolution if 1.) they assume it is fast and 2.) they track the insoluble components of the oil.

18. Page 5, Line 23. The flow rate provided is the flow rate of crude oil (dead oil), neglecting the release of light compounds (e.g., gas). The sentence reads as if the 0.1 m$^3$/s is the *in situ* flow rate of oil and liquid petroleum at the well head. The meaning of the flow rate needs to be very carefully defined. Since the model neglects dissolution, I believe this is the correct flow rate to use at the seafloor. I am just asking the authors to carefully define it—it is the flow rate of dead (black) oil, neglecting gases and highly-volatile compounds.

19. Page 5, Line 24. "Although..." I think you mean "Because..."

20. Page 5, Line 28. Rather than submit a new estimate for the removal rate of oil from the surface, can the authors not cite another study? Does the *Oil Budget Calculator* (OBC) contain this estimate? Many subsequent papers have confirmed the estimates in the OBC.

21. Page 5, Line 31. Define "super-particles". I know what they are, but they are not defined as far as I can tell in the paper.

22. Page 5, Line 32. What is the droplet size used for the particles seeded at the seafloor? Only the Li17 equation can predict droplet sizes from a jet discharge,

but technically, the Li17 equation is two equations: one for a jet and one for surface oil. Did the authors use both? Later, results are compared between Li17 and DS88. For the particles released at the seafloor, this comparison should be meaningless as DS88 does not provide estimates for a jet. As such, this comparison becomes difficult to assess.

23. Page 6, Line 5. One would not expect much difference among the models since they are both calibrated to the same experimental data. Perhaps the authors can explain a hypothesis why these models might be expected to give different results? For instance, though both may predict the same mean droplet sizes, they use different distributions. If the fate of oil in the tails of the distributions is different, then maybe one would expect differences. But, since Li17 is calibrated to the same data as DS88, I would not expect a difference *a priori*. And, if differences are only seen in the tails of the distributions, one should consider whether this involves much mass.

24. Page 6, Line 8-12. Both the DS88 and Li17 models are steady-state. I cannot understand how they could give different results after 1 hour to their results after 24 hours. The only way their results can change is if the ocean forcing changes, which may be happening, or if they are used iteratively with changing surface oil conditions. However, the text completely obscures this fact and implies that the models are unsteady, producing time-variable results. Li17 and DS88 are not and do not. This must have to do with the way OpenOil uses these in a real, time-evolving simulation. In that case, more details of OpenOil are needed, and because this is a critical conclusion of the paper, a citation to OpenOil would not be adequate: the key mechanisms / methods used to generate time-variable droplet size distributions from steady-state equations must be explained.

25. Page 6, Line 15. Small particles will be easily entrained into the ocean interior and away from the surface. The text here says the small particles have probably

"been subjected to more wind action in the last 12 hours." This sentence needs explanation. What affect of the wind? Do the authors mean the particles are smaller because the wind generated larger waves, yielding more energy, resulting in smaller particles? It does not sound like it. The paragraph reads to me that the authors are claiming the smaller particles experience greater wind drift. I would expect larger particles to experience greater wind drift as they will be closer to the surface. Please edit for clarity.

26. Page 6, Line 25ff. These results are meaningful in so far as the oil droplet sizes are understood. The above comments need significant clarification for the oil droplet sizes before this section can be rigorously evaluated.

27. Page 8, Line 5. The conclusion is that the droplet size distribution has a significant effect on horizontal and vertical distributions and a wind speed range of 5-7 m/s is noted. This appears contrary to the results, which show two different size distribution models yield very similar results. Perhaps the authors mean that the fate of the oil is linked to the size distribution and they are not claiming the two models give different results. While I also agree size distribution is critical, this paper does not show that result; instead, it shows that two reasonable models give similar results. To conclude that results would be much different with a different size distribution, additional simulations with a different size distribution would be needed.

28. I also note that this paper ignores all of the prior work using far-field Lagrangian Particle Tracking models by Elizabeth North's and Claire Paris's research groups. While North generally uses SABGOM circulation forcing, Paris also uses GoM-HYCOM. This work should be summarized and critically reviewed in the Introduction. Key results of this work that confirm or deny similar conclusions in North's or Paris's papers should also be highlighted throughout the Results and Discussion.

---

## Author Comment (AC1) · 20 Feb 2019

We thank the reviewer for very thorough comments to our manuscript. We do agree that the initialization of droplet size distribution needs to be described better and we have now prepared an improved description. In the manus we already describe that we release oil particles in a polygon at the surface (from satellite observations) + a point source at the seafloor. The release rates are quantified on p5, but we now emphasize more details on how the model is initialized.

How droplet sizes for the surface release are described in OpenOil, is documented in a very recent paper in OS (Röhrs et al. 2018). For the bottom release, we have

unfortunately overseen to describe how particle distributions are initialized, and would like to thank the reviewer for spotting this shortcoming. We have now added the following paragraph to Section 3 to clarify how droplet sizes are given: "The oil elements released at the surface are assigned random droplet radii at each entrainment incident, according to the parameterisation of size distributions from respectively DS88 or Li17, see Röhrs et al. (2018) for details. For oil elements released at the seafloor (wallhead), we use a simplistic and pragmatic approach of prescribing random radii in the range 0.5mm to 5mm, as suggested by Johansen et al. (2000). Oil elements at the sea surface (slick) are not considered to have a radius." We now also refer properly to North et al. (2011), Barker (2011) and Boufadel et al. (2014) in the introduction and discussion to compare our results with previous work.

Due to repeated surfacing and submersion from wave breaking, the particle distributions from both surface and bottom releases develop over time, and we show that the DS88 and Li17 formulations give similar size distributions after 24 hours with the same peak in the spectra. We believe that this is a new and interesting result that highlights the importance of the cycle of wave entrainment, vertical mixing and resurfacing in oil spill modeling, and we are now trying to state this even clearer..

We already have a quantitative description in the manuscript as the release rates at the surface and seafloor are described in section 3, and the time evolution of mass of oil at the surface and in the water column, at the surface, stranded and evaporated is shown in Fig 4. We focus now on a more quantitative presentation of the model results for the revised manuscript. At the same time, we think it is of interest to the reader to discuss fractions of oil stranded, at the surface etc.

Response to minor comments: **Page 1, line 9: in Abstract, "The oil droplet sizes are also relevant for the biological impact."** Changed to "oil droplet size is".

**Page 2, line 1: "significant quantities"** Not clear to us what is meant. We give two references with quantities of oil released.

**Page 2, line 22, "all important factors", I would use "major factors"** We agree. Corrected.

**Page 2, line 26, "significant effect"** Not clear what is meant here.

**Page 3, line 11, "Bleck (2002)" -> (Bleck, 2002)** Corrected.

**Page 3, line 15, "USG" -> "USGS"** Corrected

**Page 3, line 17, missing ")"** Corrected

**Page 4, line 16, "model physics", be specific.** Much more details are now provided about the model physics.

**Page 5, line 14, "state-of-the-art"** Corrected.

**Page 6, line 4, "Fig. 7"?** Numbering of figures is made automatically in latex. The order of appearance has now been corrected.

**Page 6, line 5, "former"?** Corrected to Li17.

**Page 6, line 7, ms-3 -> ms-1** Corrected

**Page 6, line 7, define "super particles"** We now just write particles throughout.

**Page 6, line 10, "very similar volume distribution". I do not agree.** Corrected to " similar volume distribution". They do have the same peak in the volume distrbution after 24 hrs.

**Page 6, line 11, "more particles", particle numbers or volume?** Corrected to marginally more oil.

**Page 6, line15, it will be interesting to check how different sizes of oils move with wind, especially those submerged in the water.** The horizontal distribution partly caused by the wind is shown in Fig. 6. The submerged particles are not moved by the wind.

**Page 8, line 3, "realistic spill rate", be quantitative.** We are quantitative and the spill rate is given in section 3. Not sure what is meant here.

**Page 8, line 11-14. Not sure what the authors wanted to deliver here. Discussion of the oil droplet sizes is not adequate. The discussion is very qualitative.** Much more detail on oil droplet size distribution is now provided with several new subsections.

**Page 8, line 30, "large surface" -> large surface area?** Corrected to "large surface area".

**Page 9, line 3-5, a few hyperlinks seem not to work.** Not sure what is meant. We do not see any hyperlinks on line 3-5. Several new superlinks are now included in the text.

**Page 9, line 10-11, be accurate describing coordinate.** Not sure what is meant. Coordinates are given.

**I would delete the last paragraph.** We agree. This is done now.

**Without the information of how oil sizes were calculated and what oil fate and transport model was used, I think this manuscript is lack of fundamental information to justify the rationale of the simulation.** We have a long paragraph describing the oil transport model used, and several references are given. The ADIOS Oil library is also well described in references. We have added several new subsections describing how oil droplet size distribution is calculated.

Please also note the supplement to this comment:
https://www.ocean-sci-discuss.net/os-2018-130/os-2018-130-AC1-supplement.pdf

**Supplement:**

**Revisiting the DeepWater Horizon spill: High resolution model** simulations of effects of oil droplet size distribution and river fronts**

Lars R Hole1, Knut-Frode Dagestad1, Johannes Röhrs1, Cecilie Wettre1, Vassiliki H. Kourafalou2, Ioannis Androulidakis2, Matthieu Le Hénaff3,4, Heesook Kang2, and Oscar Garcia-Pineda5

1Norwegian Meteorological Institute, Allegt. 70, 5007 Bergen, Norway
 2University of Miami, Rosenstiel School of Marine and Atmospheric Science, Miami, FL, USA
 3University of Miami, Cooperative Institute for Marine and Atmospheric Studies, Miami, FL, USA
 4NOAA Atlantic Oceanographic and Meteorological Laboratory, Miami, FL, USA
 5WaterMapping, Gulf Breeze, FL, USA

Correspondence: Lars Robert Hole (lrh@met.no)

**Abstract.** An open source ocean trajectory framework, OpenDrift, is used to simulate the 2010 DeepWater Horizon oil spill. Metocean forcing data are taken from the GoM-HYCOM 1/50° ocean model with realistic river input and ECMWF global forecast products of wind and wave parameters with 1/8° resolution. OpenDrift includes the integrated oil drift module OpenOil, which includes a number of relevant processes, such as emulsification, wave entrainment, and droplet formation. This takes

- 5 account of the actual oil type/properties, using the ADIOS oil weathering database of NOAA. The effect of using a newly developed parameterization for oil droplet size distribution is studied, compared to a traditional algorithm. Although the algorithms provide different distributions for a single wave breaking event, it is found that the net difference after long time simulations is negligible, indicating that the outcome is robust regarding the choice of parameterization. In both cases, the size of the droplets controls how much oil is present at the surface and hence are subject to wind and Stokes drift. The oil droplet
- 10 size is also relevant for the biological impact. Next, the effect of removing river outflow in the ocean model is investigated in order to showcase effects of river induced fronts on oil spreading. A consistent effect on the amount and location of stranded oil is found, and considerable impact of river induced fronts is seen on the location of the surface oil patch. During a case with large river outflow (May 20-27, 2010), the total amount of stranded oil is reduced by about 50% in the simulation with no river input.

15 Copyright statement. TEXT

**1 Introduction**

The presence of both shelf and open sea dynamics makes the Northern Gulf of Mexico (NGoM) a topographically and dynamically complex and interesting study area, in the presence of intense oil exploration. Interactions of the Mississippi River (MR) plume and the Loop Current (LC) system were found to be important for the transport and fate of oil during the 2010 DeepWater Horizon (DWH) incident (Kourafalou and Androulidakis, 2013; Le Hénaff et al., 2012).

According to the U.S. Energy Information Administration, more than 45% of total U.S. petroleum refining capacity and 51% of total natural gas processing plant capacity is located along the Gulf coast www.eia.gov/special/gulf of mexico/. Oil

5 leaks and accidents, such as the explosion on the DWH platform in 2010 (at 28.737°N, 88.366°W), have released significant quantities of hydrocarbons (Crone and Tolstoy (2010); McNutt et al. (2011)) in the sensitive marine environment around the MR Delta, and over the LouisianA TEXas (LATEX) and Mississippi Alabama FLoridA (MAFLA) shelves (Kourafalou and Androulidakis, 2013) (Fig. 1).

Many studies have dealt with simulation of the DWH spill (North et al., 2011; Mariano et al., 2011; MacFadyen et al.,

- 10 2011; Barker, 2011; Le Hénaff et al., 2012; Paris et al., 2012; Kourafalou and Androulidakis, 2013), with focus on both subsurface (Paris et al., 2012) and surface (Le Hénaff et al., 2012) transport. North et al. (2011) used a plume model to predict a stratification-dominated near field, in which small oil droplets detrained from the central plume containing faster rising large oil droplets and gas bubbles and became trapped by density stratification. They showed that simulated droplets with diameters between 10 and 50 µm formed a distinct subsurface plume, which was transported horizontally and remained in the subsurface
- 15 for >1 month. In contrast, droplets with diameters >90  $\mu m$  rose rapidly to the surface. Le Hénaff et al. (2012) focused on oil transport on the water surface and found that the wind played a major role in advecting the oil to the northern GOM. Barker (2011) conducted Monte Carlo simulations consisting of 500 individual oil trajectory scenarios using historical data of water currents and winds. The results by Barker (2011) indicated that, in approximately 75% of the scenarios, oil would be transported out of the GOM by the Loop Current. This means that the actual trajectory of oil from the DWH falls in the 25%

20 of scenarios.

Androulidakis et al. (2018b) carried out a field experiment deploying surface drifters at different times near the Taylor Energy Site which is located in the vicinity of the MR outflow region over the NGoM and near the DWH site (approximately at 28.938°N, 88.978°W, Fig. 1). This multi-platform observational experiment was conducted in April 2017 to investigate the main transport pathways from the Taylor Site and toward the NGoM continental shelves and offshore, toward the Gulf interior.

- 25 Results indicated that the surface transport was determined by the MR plume extension over the Taylor Energy Site and the river induced fronts in combination with local circulation, prevailing winds and broader regional dynamics (LC system). The drifters deployed during the field experiment in tandem with satellite data, drone imagery, wind measurements, and marine radar derived currents and images described three major transport pathways, in agreement with the three major circulation patterns of the MR plume (Schiller and Kourafalou, 2010; Androulidakis et al., 2018a; Schiller et al., 2011; Schiller and Kourafalou, 2010; Androulidakis et al., 2018a; Schiller et al., 2011; Schiller and Kourafalou, 2010; Androulidakis et al., 2018a; Schiller et al., 2011; Schiller and Kourafalou, 2010; Androulidakis et al., 2018a; Schiller et al., 2011; Schiller and Kourafalou, 2010; Androulidakis et al., 2018a; Schiller et al., 2011; Schiller and Kourafalou, 2010; Androulidakis et al., 2018a; Schiller et al., 2011; Schiller and Kourafalou, 2010; Androulidakis et al., 2018a; Schiller et al., 2011; Schiller and Kourafalou, 2010; Androulidakis et al., 2018a; Schiller et al., 2011; Schiller and Kourafalou, 2010; Androulidakis et al., 2018a; Schiller et al., 2011; Schiller and Kourafalou, 2010; Androulidakis et al., 2018a; Schiller et al., 2011; Schiller and Kourafalou, 2010; Androulidakis et al., 2018a; Schiller et al., 2011; Schiller and Kourafalou, 2010; Androulidakis et al., 2018a; Schiller et al., 2011; Schiller and Kourafalou, 2010; Androulidakis et al., 2014; Schiller and Schiller et al., 2014; Schiller et al., 20
- 30 Kourafalou, 2014).

The drifters deployed by Androulidakis et al. (2018b) followed the two prevailing coastal currents associated with MR plume dynamics (downstream/upstream moving westward/eastward of the Mississippi Delta) and an offshore pathway under the influence of basin-wide circulation. Near the Taylor site, the existence of multiple river fronts influence the fate of oiled waters, preventing the transport of hydrocarbon toward the delta like a natural boom barrier, trapping and directing the oil

35 either westward or eastward in agreement with Kourafalou and Androulidakis (2013), who showed a similar interaction during

the DWH accident. *In situ* thermohaline measurements around the Taylor Energy Site and across the river front showed that the MR plume near the Taylor Site was 5m to 10m deep, while the clearer ocean water column was characterized by a 40 m upper-ocean homogeneous layer, mainly controlled by temperature.

- In this study, an open source Lagrangian oil drift model, OpenOil, has been used to simulate the DWH oil spill evolution. 5 OpenOil takes into account major factors that influence the short term drift of an surface oil slick such as metocean forcing (including Stokes drift), emulsification, evaporation and vertical entrainment and mixing. Dissolution, which is important for subsurface oil spills, is not yet implemented. Simulations are initiated from satellite observations and a point source at the sea floor. The effect of using two different oil droplet size distribution on the horizontal drift and vertical mixing is discussed. Next, the study showcases how NGoM oil pathways are influenced by river plume circulation and river induced fronts. It is also
- 10 investigated whether the use of realistic daily river discharge has a significant effect on the simulated location of the Surface Oil Patch (SOP) and stranding of oil.

**2 Methods and data**

**2.1 Shapefiles of surface oil patch**

Shape files derived from satellite analysis of the DWH SOP can be accessed through the NOAA-ERMA website erma.noaa.
gov/gulfofmexico. In the present study, oil particles were seeded uniformly within the region enveloping the thick and thin oil slicks with no distinction. The shapefiles were used here for both initialization of the oil drift simulations and for verification of results.

**2.2 Metocean forcing**

In the cases presented here, the ocean circulation fields come from a data-assimilative, high-resolution (1/50°, 1.8 km) config-20 uration of the Hybrid Coordinate Ocean Model (HYCOM - www.hycom.org) in the Gulf of Mexico (GoM), developed by the authors. This configuration, which we refer to as GoM-HYCOM 1/50°, uses daily river forcing and data assimilation (GoM-HYCOM 1/50°). The HYCOM model has a flexible, hybrid vertical coordinate system, in which the distribution of vertical layers is optimized: they are isopycnal in stratified water columns, sigma terrain-following in regions with sharp topography, and isobaric in the mixed layer and very shallow areas (Bleck, 2002). More information about the HYCOM model is available

- 25 in the model user's manual www.hycom.org and the references therein. The GoM-HYCOM 1/50° covers the entire GoM and uses 32 vertical levels. The model configuration is similar to the one used by Le Hénaff and Kourafalou (2016), with the real-istic river forcing parameterization developed by Schiller and Kourafalou (2010). The river discharge data were obtained from the Army Corps of Engineers and the U.S. Geological Survey (USGS www.usgs.gov). The model is initialized in October 2009 with fields from the operational Global HYCOM (GLB-HYCOM) simulation run at the Naval Research Laboratory at the
- 30 Stennis Space Center (GLB-HYCOM expt\_90.8, www.hycom.org), and it is nested at the open boundaries with model fields from the same simulation. The atmospheric forcing is based on the 3-hourly winds, thermal forcing and precipitation forecast

fields from the European Centre for Medium-Range Weather Forecasts (ECMWF www.ecmwf.int), with spatial resolution of 0.125° (see below). The model assimilates satellite observations of Sea Surface Temperature and Sea Surface Height, and in situ observations of temperature and salinity from buoys, cruises, surface drifters, Argo floats and XBT casts. More details about the model configuration can be found in Le Hénaff and Kourafalou (2016) and Androulidakis et al. (2018a). For the

- 5 present study, two simulations were performed: one with the attributes mentioned above, called *Reference* simulation, and one called *No river*, in which the salinity fronts have been removed by shutting off the river discharge, setting precipitation to zero, and turning off the assimilation of salinity profiles. All the other forcing conditions (e.g. meteorological, boundary) remained the same between the two experiments in order to investigate the impact of an individual forcing mechanism (here the Mississippi buoyant discharge and the related density fronts) on the circulation features and furthermore on the oil spill extensions
- 10 during the DwH period. The outputs from both simulations are available at the Gulf of Mexico Research Initiative Information and Data Cooperative (GRIIDC data.gulfresearchinitiative.org/).

[revised manuscript text omitted]

Following Li et al. (2017b), the volume (V) droplet size spectrum is described by the median droplet diameter,  $D_{50}^V$ , as

$$D_{50}^{V} = d_{o}r \left(1 + 10 \text{Oh}\right)^{p} \cdot \text{We}^{q}$$
(1)

with the empirical coefficient r = 1.791 and the exponents p = 0.460 and q = -0.518. The PDF for the droplet size distribution follows a log-normal distribution around the medium diameter with a logarithmic base-10 standard deviation of s = 0.38 (Eq. 16 in Röhrs et al. (2018)).

The Weber number, We, is a dimensionless number describing the relative importance of inertial forces and oil-water interfacial tension. It is a function of the sea water density,  $\rho_w$ , the significant wave height,  $H_s$ , and the oil-water interfacial tension,  $\sigma_{o-w}$ , and is given by

$$We = \frac{\rho_w g H_s d_o}{\sigma_{o-w}},\tag{2}$$

where g is the acceleration of gravity and  $d_o = 4\sqrt{\frac{\sigma_{o-w}}{g(\rho_w - \rho_o)}}$  is the Rayleigh-Taylor instability maximum diameter.

The Ohnesorge number, Oh, is a dimensionless number describing the ratio of viscous forces to inertial and surface tension forces. It is a function of the dynamic oil viscosity,  $\mu_o$ , oil density,  $\rho_o$ , and oil-water interfacial tension:

30
$$\operatorname{Oh} = \frac{\mu_o}{\sqrt{(\rho_o \sigma_{o-w} d_o)}}.$$
 (3)

$$V(d) = d^{-0.6} \text{for} d_{min} < d < d_{max}$$
(4)

where d is the droplet diameter. Minumum and maximum droplet radii are set to 10e-6 and 10e-3, respectively. The exponent of -0.7 in the volume size distribution corresponds to an exponent in the number size distribution of 2.3 (Tkalich and Chan, 5 2002).

Droplet sizes are assigned to oil particles each time a particle is submerged by breaking waves, following the wave entrainment algorithm of (Li et al., 2017c). The implementation of this algorithm in OpenOil is described if full detail in (Röhrs et al., 2018). The droplet sizes for individual particles are drawn from a random distribution according to the chosen size distribution. The size distributions represent conditions for a stochastic wave entrainment event, representing equilibrium conditions

10 during a model time step. It is noted that the overall size distribution of all submerged oil in the simulation is further subject to changes, as weather conditions and the oil's emulsification rate change and oil droplets of various sizes are subject to various resurfacing time scales. Resurfaced particles are considered to be part of a surface slick, and are assigned a new droplet size once they are re-entrained. Oil elements at the sea surface (slick) are not considered to have a radius.

**Droplet size distribution during deep blowouts**

For oil elements released at the seafloor (wellhead), a simplistic and pragmatic approach of prescribing random radii in the 15 range 0.5 mm to 5 mm was used, as suggested by Johansen (2000).

**Horizontal transport**

With regard to horizontal drift, three processes are considered: Any element, whether submerged or at the surface, drifts along with the ocean current. Elements are further subject to Stokes drift corresponding to their actual depth. Surface Stokes drift is 20 normally obtained from a wave model, and its decline with depth is calculated as described in Breivik et al. (2016). Oil elements at the ocean surface are additionally moved with a factor of 2% of the wind. Together with the Stokes drift (typically 1.5% of the wind at the surface), this sums up to the commonly found empirical value of 3.5% of the wind speed (Schwartzberg, 1971). The magnitude of the wind drift factor is discussed in Jones et al. (2016) who stated that a 2% wind drift factor was required in OpenOil to reproduce their observations of a SOP in the North sea. In essence, this is believed to be a compensation factor

25

for the inability of any ocean model to represent the strong shear current in the upper few centimeters/decimeters of the ocean, and not surface oil actually moving relative to the water.

The three horizontal drift components may lead to a very strong gradient of drift magnitude and direction in the upper few meters of the ocean. For this reason, it is also of critical importance to have a good description of the vertical oil transport processes.

**Vertical transport**

Oil elements at the surface, regarded as being in the state of an oil slick, may be entrained into the ocean by breaking waves. The entrainment of oil droplets depends on both the wind and wave (breaking) conditions, but also on the oil properties, such as viscosity, density and oil-water inter facial tension. The buoyancy of droplets is calculated according to empirical relationships

5 and the Stokes law following Tkalich and Chan (2002), dependent on ocean stratification based on temperature and salinity from the ocean model, and the viscosities and densities of oil and water.

In addition to the wave induced entrainment, the oil elements are also subject to vertical turbulence throughout the water column, described using a random-walk scheme based on the turbulent eddy diffusivity from the ocean model.

**Weathering**

10 In order to calculate weathering of the oil, OpenOil interfaces with the open source ADIOS oil library, developed by NOAA (github.com/NOAA-ORR-ERD/OilLibrary) and (Lehr et al., 2002). In addition to state-of-the-art parametrization of weathering processes such as evaporation, emulsification and dispersion, this software contains a database of measured properties of almost 1000 oil types from around the world. As oils from different sources or wells have vastly different properties, such a database is of vital importance for accurate results. The ADIOS oil library is also used by the NOAA oil drift model

15 github.com/NOAA-ORR-ERD/PyGnome.

The weathering algorithms describes evaporation and emulsification rate of oil, i.e. the water content. The emulsification and evaporation greatly affect oil density, viscosity and oil-water interfacial tension, and thereby the droplet size distribution through Eqs. 1-3. OpenOil take into consideration weathering processes that are dominating in the initial oil spill period of 2-3 days. Long-term weathering processes such as sedimentation and microbial degradation are not considered in this study.

**20 3 Results**

A first set of simulations are carried out to investigate the effect of oil droplet size distribution. A second set focuses on the effect of river induced fronts. According to Crone and Tolstoy (2010), the average flow rate from the oil well between 22 April and 3 June 2010 was estimated to 0.1 m3 sec-1, assuming a liquid oil fraction of 0.4. Gas and highly volatile compounds are not considered here. After the riser was removed and until the leak was sealed on 15 July, the flow rate increased to 0.12 m3 sec-1, corresponding to 10,368 m3 day-1. An assumption was made on of how much oil was present at the sea surface at the initialization of the simulation. The residence of oil at the sea surface depends heavily on oil properties as well as environmental conditions such as temperature of ocean and air, wind and waves, as described above. According to our mass balance calculations for the DWH spill, using the Light Louisiana Sweet oil type from the NOAA oil library and environmental conditions as described above, it seems reasonable to assume that 80% of the oil mass is removed from the surface after 10

days. This is within the range in our simulations that is typically 60 to 95% (see examples of mass balance plots further down).While Boufadel et al. (2014) assumed a constant removal rate of surface oil 20% per day, the removal rate in the present

simulation and in reality will vary with wind and wave conditions. The simulations for May 2010 are initialized by seeding 48730 particles in a polygon obtained from NOAA shapefiles. Each particle represents initially 1 m3 oil. A continuous point source at the sea floor seeds an additional 8460 particles (8460 m-3) per day during the simulation. After June 3rd these numbers are increased by 20% to 10368 m3 day-1. The oil elements released at the surface are assigned droplet radii at each entrainment

5 incident, according to the parameterisation of size distributions from respectively DS88 or Li17, see Röhrs et al. (2018) for details. Oil elements at the sea surface (slick) are not considered to have a radius.

Around 20-25 May 2010 there was a significant outflow of the Mississippi River (Fig. 2) and part of the SOP was entrained along the LC resulting in a formation popularly referred to as the "tiger tail" (Fig. 3). The OpenOil simulation shown on top in Fig. 3 is carried out using the classical DS88 oil droplet size distribution (Delvigne and Sweeney, 1988). Fig. 3 lower panel

- 10 shows the results from repeating this simulation using the new Li17 formulation. In Fig. 4 the mass balance during seven days for the Li17 simulation is shown. There is virtually no difference between DS88 and Li17 and only Li17 is shown here. Both formulations result in about the same fraction of oil at the surface (about 50%) after 7 days, with moderate wind speeds of up to 7.2ms-1. The light compounds evaporate fast after release hence the more heavy compounds are tracked here ("dead oil"). Patches of thick oil (where the particles retain nearly 100% of their mass) are visible over a larger area (Fig. 3). Larger oil
- 15 droplets will rise faster to the surface (North et al., 2011; Röhrs et al., 2018), and DS88 provides a much higher fraction of large droplet after one hour compared to Li17 (Fig. 5, upper panels). However, it turns out that the DS88 and Li17 provide similar volume distributions after a 24 hr test simulation (Fig. 5, lower panels). Still, the peak in the distribution is at around  $100 \ \mu m$  for both formulations, and rapid rising to the surface can be expected according to North et al. (2011). Due to the small difference between DS88 and Li17, the DS88 simulation results in just marginally more oil stranded after seven days (13.2
- 20 vs 12.8 %), particularly west of the Mississippi Delta (Figs. 3). A higher fraction of oil at the surface provides more efficient transport by wind and waves towards the shore and larger likelihood of stranding. For both simulations, the oil particles quickly loose 40-50% of their mass, mostly through evaporation.

Fig. 6 shows the geographical distribution of particle diameters at the surface. It is apparent that smaller particles are present outside the MAFLA shelf, probably because the oil particles have been subject to more wind and hence wave action and natural dispersion in the last 24 hours (Fig. 7).

[revised manuscript text omitted]

To the best of our knowledge, this is the first time the importance of the effect of river fronts on oil slick transport in the Gulf of Mexico has been demonstrated using high resolution forcing and a fully fledged oil drift model.

**TEXT**

25

Code availability. github.com/OpenDrift/opendrift

Data availability. TEXT

Code and data availability. TEXT

11

Sample availability. TEXT

Video supplement. TEXT

**Appendix A**

A1

5 Author contributions. TEXT

Competing interests. TEXT

Disclaimer. TEXT

Acknowledgements. This research was made possible by a grant from The Gulf of Mexico Research Initiative (award "Influence of river induced fronts on hydrocarbon transport," GOMA 23160700). Atmospheric and wave data were kindly provided by the European Center
for Medium-Range Weather Forecasts (ECMWF). M. Le Hénaff acknowledges partial support from the Physical Oceanography Division at NOAA's Atlantic Oceanographic and Meteorological Laboratory, AOML. The outputs from the HYCOM simulations are available at the Gulf of Mexico Research Initiative Information and Data Cooperative (GRIIDC - doi:10.7266/N7NG4NPC).

**Figure 1.** Map of the Northern Gulf of Mexico showing the geographical locations mentioned in the text. The map insert shows the Mississippi River (MR) delta in blue with the three major river passes that release MR water into the Gulf.

**Figure 2.** Discharge from Mississippi River in the Northern Gulf of Mexico during late spring and early summer 2010. Data kindly provided by U.S. Geological Survey (USGS www.usgs.gov) and U.S. Army Corps of Engineers

---

## Author Comment (AC2) · 20 Feb 2019

[12pt]article

We would like to thank this reviewer for taking the time to provide very detailed and specific comments to our manuscript. Below are our responses with the reviewers comments in bold letters.

**The article is too wordy. For example, the Introduction seems to focus on discussing a couple papers related to the MR rather than providing a background for investigating the DWH spill in relation to existing works that modeled the DWH**

[Figure]

surface spill, such as the works of McFaddyen et al. (EOS, 2011) and Boufadel et al. (ES&T, 2014). **In fact, only by comparing to such works that the authors can highlight their contributions. How do the results based on the more accurate HYCOM compare to results obtained using NCOM, NGOM, etc? The oil behavior appears to be an aftermath of the exercise, which is to highlight the role of the MR. One is led to wonder, what would be the fate of the oil if one assumes that the droplets are neutrally buoyant? What was the depth of mixing? In essence, there were too many things that the authors considered giving the impression that the article was put together hastily without a goal in mind. Detailed comments are below.** We thank the reviewer for pointing this out. The introduction is now rewritten and with additional references to relevant studies.

**"Next, the study showcases how NGoM oil pathways are influenced by river plume circulation and river induced fronts. We also investigate whether" It is preferable to stick to one tone of writing, either passive (the study discusses) or active (we investigate). Mixing the two is not common.** We agree. We have now used passive tense throughout.

**"As part of the efforts made by the National Oceanic and Atmospheric Administration (NOAA) to assess the extent and impact of the DWH spill, participants on this team analyzed hundreds of satellite images (microwave and optical) and produced oil extent delineations throughout the lifetime of the spill. Classifications derived from the satellite analysis of the DWH SOP can be accessed through the NOAA-ERMA website (ERM)." Which team? The NOAA team or the authors? Also, why not writing that "we analyzed satellite images (microwave and optical) of the spill from the NOAA site ERMA" ? Why using such a wordy approach?** We agree. We have made this description much shorter and to the point.

**"The shapefiles were used for both initialization of the oil drift simulations and for verification of results". By whom? By NOAA or by the current authors?** By the currents authors. We have reformulated to make it clear.

**"For the present study, we performed two simulations: one with the attributes men- tioned above, called Reference simulation, and one called No river, in which the salin- ity fronts have been removed by shutting off the river discharge, setting precipitation to zero, and turning off the assimilation of salinity profiles." Please mention here in a sentence why the two choices were made. Assuming that one wants to evaluate the impact of the MR, how realistic would be to turn off the flow of the Mississippi? Wouldn't using the minimum flow be more realistic? Is the goal to conduct an academic exercise to evaluate the impact of the MR on the near shore hy- drodynamics? If yes, then, this should be conducted only in a Monte Carlo framework as done by NOAA's Barker (EOS, 2011). Otherwise, the results and conclusions would be dependent on the hydrodynamic and climatological conditions during the DWH spill.**

We followed a standard methodology that is traditionally followed by model simulations (often called "twin experiments") but also by theoretical approaches (as in examining prevailing balance of forces in momentum equations by making assumptions that re- move one or more equation terms). To fully evaluate the impact of an individual forcing mechanism (here the Mississippi buoyant discharge), this forcing has to be completely shut off. Same for the impact of additional forcings influencing salinity variability (pre- cipitation in particular). All other forcings for the DwH period remain the same. The intent is exactly to keep all other conditions similar, especially the wind-driven flows and the offshore circulation (Loop Current and eddies). Keeping any salinity gradients (like using a small discharge rate etc.) would prevent these experiments from giving conclusive results.

**"The physical mechanism behind this wind drift factor is not obvious, and is discussed in Jones et al. (2016)." What was the summary from Jones et al. (2016)? The increase in oil drift is well understood for anyone who worked on transport. Based on simple momentum considerations, one could learn that anything that is on the top of the water surface (and has a smaller density) would**

**move faster than the water beneath it if the wind is blowing (i.e., the whole idea of a sail boat !). Also, the fact that the oil viscosity is much higher than that of the water beneath it, makes the oil behaves locally as an "object". But if the authors have no idea why the oil moves faster than the water beneath it, I am not sure they should be conducting oil spill simulations using complicated models for evaporation, emulsification, dispersion, etc. Also, these processes were mentioned early on, and then were not discussed afterwards.** As discussed in Jones et al. (2016), the wind drift factor need not be solely the movement of oil relative to water, but may be partly a compensation factor for the inability of any ocean model to represent the strong shear current in the upper few centimeters/decimeters of the ocean, or biases due to the truncated spectral tail of the wavemodel-derived Stokes drift, as well as unresolved Langmuir jet currents. We have updated the text to make this clearer. Emulsification and dispersion is related to oil droplet size and is hence discussed in the paper. Evaporation is shown in the mass balance plot (Fig. 4) and also mentioned in the discussion.

**"According to our mass balance calculations for the DWH spill, using the Light Louisiana Sweet oil type from the NOAA oil library and environmental conditions as described above, it seems reasonable to assume that 80% of the oil mass is removed from the surface after 10 days. This is within the range in our simulations that is typically 60 to 95% (see examples of mass balance plots further down). The simulations for May 2010 are initialized by seeding 48730 super-particles in a polygon obtained from NOAA shapefiles. Each particle represents initially 1 m3 oil. A continuous point source at the sea floor seeds an additional 8460 particles (8460 m-3) per day dur- ing the simulation. After June 3rd these numbers are increased by 20% to 10368 m3 day-1." The work by Boufadel et al. (2014, EST) specifically addressed this issue. See, for example, their Figure 2. They found that the oil disappears of the surface at around 20% per day.**

We thank the reviewer for pointing out this publication. However, Boufadel et al used a

constant removal rate, while we use a removal rate depending on wind and waves and the specific oil composition, as given by the Adios oil library. As the reviewer probably know, the residence time of the oil at the surface is highly variable and dependent of oil type and wind speed. Very few field observations are available (a new study is in prep by some of the current authors). We believe that 20% remaining oil after 10 days is representative for the conditions during our simulations. This estimate is only used to decide the amount of oil to be seeded in the polygon at the first time step.

**" The OpenOil simulation shown on top in Fig. 3 is carried out using the classical DS88 oil droplet size distribution (Delvigne and Sweeney, 1988)." What was the thickness of the "Surface Layer" in the simulation? 1.0 m? 10.0 m? Obviously, this affects the meaning of concentration.** The top layer in the ocean model is 1m. This is now pointed out the the manuscript.

**"Fig. 3 lower panel shows the results from repeating this simulation using the new Li17 formulation. In Fig. 7 we show the mass balance during seven days for 5 the Li17 simulation. There is virtually no difference between DS88 and Li17 and only the former is shown here."Why discussing Figure SEVEN prior to Figures 4, 5, and 6? Should Figure 7 be made as Figure 4?** We are sorry, the figure numbering is carried out automatically by the Latex software. The order of appearance has been corrected now.

**Table 2 provides oil deposited on the shorelines, and the amount needs to be compared with the work of Boufadel et al. (2014, ES&T).** Our results are not directly comparable to Boufadel et al. (2014) since they looked at different periods than us and the used constant daily removal rates from the surface (0,10%and 20%). We are using a varying removal rate depending on wind speed and wave breaking as described in our manus and by Röhrs et al (2018).

**How are the authors deciding what is on the surface and what is submerged? What was the cutoff depth? I find it unusual that they did not provide such in-**

**formation considering that GOMRI works including their own showed the impor-
tance of depth (on the order of centimeters) on the hydrodynamics.** Every particle
has an individual, varying depth for each time step as resulting from buoyant uplift and
turbulent mixing. We have no cutoff depth since we have a point source at the sea
floor.

**"A realistic description of droplet formation is required to describe the effects of
an oil spill on the environment" True, but there is no citation for this statement.**
We now refer to Boufadel et al. (2014) and North et al. (2011).

**"The parts of the oil spill at the surface is more hazardous to birds and the beach
communities, while the small, submerged parts will have a substantially larger
surface to 30 interact with water and plankton (Carroll et al., 2018)." The authors
cite a paper on the impact to the Northeast fisheries in spite of the HUGE amount
of papers dealing with the DWH spill. It is pretty strange. Maybe they can start
with the review by Short (2017, Archives of Environmental Contamination and
Toxicology). Maybe they can peruse the GOMRI website?** We now refer to Short
(2017)

**"To the best of our knowledge, this is the first time the importance of the effect
of river fronts on oil slick transport in the gulf has been demonstrated using high
resolution models." The work of the authors themselves addressed the impact
of the MR on coastal hydrodynamics. Kourafalou, V. H. and Androulidakis, Y. S.:
Influence of Mississippi River induced circulation on the Deepwater Horizon oil
spill transport, Journal of Geophysical Research: Oceans, 118, 3823–3842, 2013.
Le Henaff, M. and Kourafalou, V. H.: Mississippi waters reaching South Florida
reefs under no flood conditions: synthesis of observing and modeling system
findings, Ocean Dynamics, 66, 435–459, 2016.** Kourafalou and Androulidakis (2013)
used a purely hydrodynamic model, they did not use an oil drift model. Their results
were based on model salinity (for the river plume signal) and data derived oil patches
(for the surface signature of the oil). However, we have now reformulated to "To the

best of our knowledge, this is the first time the importance of the effect of river fronts on oil slick transport in the gulf has been demonstrated using high resolution forcing and a fully fledged oil drift model."

**Supplement:**

**Revisiting the DeepWater Horizon spill: High resolution model simulations of effects of oil droplet size distribution and river fronts**

Lars R Hole[1], Knut-Frode Dagestad[1], Johannes Röhrs[1], Cecilie Wettre[1], Vassiliki H. Kourafalou[2], Ioannis Androulidakis[2], Matthieu Le Hénaff[3,4], Heesook Kang[2], and Oscar Garcia-Pineda[5]

[1]Norwegian Meteorological Institute, Allegt. 70, 5007 Bergen, Norway
[2]University of Miami, Rosenstiel School of Marine and Atmospheric Science, Miami, FL, USA
[3]University of Miami, Cooperative Institute for Marine and Atmospheric Studies, Miami, FL, USA
[4]NOAA Atlantic Oceanographic and Meteorological Laboratory, Miami, FL, USA
[5]WaterMapping, Gulf Breeze, FL, USA

**Correspondence:** Lars Robert Hole (lrh@met.no)

**Abstract.** An open source ocean trajectory framework, OpenDrift, is used to simulate the 2010 DeepWater Horizon oil spill. Metocean forcing data are taken from the GoM-HYCOM 1/50º ocean model with realistic river input and ECMWF global forecast products of wind and wave parameters with 1/8º resolution. OpenDrift includes the integrated oil drift module OpenOil, which includes a number of relevant processes, such as emulsification, wave entrainment, and droplet formation. This takes account of the actual oil type/properties, using the ADIOS oil weathering database of NOAA. The effect of using a newly developed parameterization for oil droplet size distribution is studied, compared to a traditional algorithm. Although the algorithms provide different distributions for a single wave breaking event, it is found that the net difference after long time simulations is negligible, indicating that the outcome is robust regarding the choice of parameterization. In both cases, the size of the droplets controls how much oil is present at the surface and hence are subject to wind and Stokes drift. The oil droplet size is also relevant for the biological impact. Next, the effect of removing river outflow in the ocean model is investigated in order to showcase effects of river induced fronts on oil spreading. A consistent effect on the amount and location of stranded oil is found, and considerable impact of river induced fronts is seen on the location of the surface oil patch. During a case with large river outflow (May 20-27, 2010), the total amount of stranded oil is reduced by about 50% in the simulation with no river input.

*Copyright statement.* TEXT

**1 Introduction**

The presence of both shelf and open sea dynamics makes the Northern Gulf of Mexico (NGoM) a topographically and dynamically complex and interesting study area, in the presence of intense oil exploration. Interactions of the Mississippi River

(MR) plume and the Loop Current (LC) system were found to be important for the transport and fate of oil during the 2010 DeepWater Horizon (DWH) incident (Kourafalou and Androulidakis, 2013; Le Hénaff et al., 2012).

According to the U.S. Energy Information Administration, more than 45% of total U.S. petroleum refining capacity and 51% of total natural gas processing plant capacity is located along the Gulf coast www.eia.gov/special/gulf_of_mexico/. Oil leaks and accidents, such as the explosion on the DWH platform in 2010 (at 28.737ºN, 88.366ºW), have released significant quantities of hydrocarbons (Crone and Tolstoy (2010); McNutt et al. (2011)) in the sensitive marine environment around the MR Delta, and over the LouisianA TEXas (LATEX) and Mississippi Alabama FLoridA (MAFLA) shelves (Kourafalou and Androulidakis, 2013) (Fig. 1).

Many studies have dealt with simulation of the DWH spill (North et al., 2011; Mariano et al., 2011; MacFadyen et al., 2011; Barker, 2011; Le Hénaff et al., 2012; Paris et al., 2012; Kourafalou and Androulidakis, 2013), with focus on both subsurface (Paris et al., 2012) and surface (Le Hénaff et al., 2012) transport. North et al. (2011) used a plume model to predict a stratification-dominated near field, in which small oil droplets detrained from the central plume containing faster rising large oil droplets and gas bubbles and became trapped by density stratification. They showed that simulated droplets with diameters between 10 and 50 $\mu m$ formed a distinct subsurface plume, which was transported horizontally and remained in the subsurface for >1 month. In contrast, droplets with diameters >90 $\mu m$ rose rapidly to the surface. Le Hénaff et al. (2012) focused on oil transport on the water surface and found that the wind played a major role in advecting the oil to the northern GOM. Barker (2011) conducted Monte Carlo simulations consisting of 500 individual oil trajectory scenarios using historical data of water currents and winds. The results by Barker (2011) indicated that, in approximately 75% of the scenarios, oil would be transported out of the GOM by the Loop Current. This means that the actual trajectory of oil from the DWH falls in the 25% of scenarios.

Androulidakis et al. (2018b) carried out a field experiment deploying surface drifters at different times near the Taylor Energy Site which is located in the vicinity of the MR outflow region over the NGoM and near the DWH site (approximately at 28.938ºN, 88.978ºW, Fig. 1). This multi-platform observational experiment was conducted in April 2017 to investigate the main transport pathways from the Taylor Site and toward the NGoM continental shelves and offshore, toward the Gulf interior. Results indicated that the surface transport was determined by the MR plume extension over the Taylor Energy Site and the river induced fronts in combination with local circulation, prevailing winds and broader regional dynamics (LC system). The drifters deployed during the field experiment in tandem with satellite data, drone imagery, wind measurements, and marine radar derived currents and images described three major transport pathways, in agreement with the three major circulation patterns of the MR plume (Schiller and Kourafalou, 2010; Androulidakis et al., 2018a; Schiller et al., 2011; Schiller and Kourafalou, 2014).

The drifters deployed by Androulidakis et al. (2018b) followed the two prevailing coastal currents associated with MR plume dynamics (downstream/upstream moving westward/eastward of the Mississippi Delta) and an offshore pathway under the influence of basin-wide circulation. Near the Taylor site, the existence of multiple river fronts influence the fate of oiled waters, preventing the transport of hydrocarbon toward the delta like a natural boom barrier, trapping and directing the oil either westward or eastward in agreement with Kourafalou and Androulidakis (2013), who showed a similar interaction during

the DWH accident. *In situ* thermohaline measurements around the Taylor Energy Site and across the river front showed that the MR plume near the Taylor Site was 5m to 10m deep, while the clearer ocean water column was characterized by a 40 m upper-ocean homogeneous layer, mainly controlled by temperature.

In this study, an open source Lagrangian oil drift model, OpenOil, has been used to simulate the DWH oil spill evolution. OpenOil takes into account major factors that influence the short term drift of an surface oil slick such as metocean forcing (including Stokes drift), emulsification, evaporation and vertical entrainment and mixing. Dissolution, which is important for subsurface oil spills, is not yet implemented. Simulations are initiated from satellite observations and a point source at the sea floor. The effect of using two different oil droplet size distribution on the horizontal drift and vertical mixing is discussed. Next, the study showcases how NGoM oil pathways are influenced by river plume circulation and river induced fronts. It is also investigated whether the use of realistic daily river discharge has a significant effect on the simulated location of the Surface Oil Patch (SOP) and stranding of oil.

**2 Methods and data**

**2.1 Shapefiles of surface oil patch**

Shape files derived from satellite analysis of the DWH SOP can be accessed through the NOAA-ERMA website erma.noaa. gov/gulfofmexico. In the present study, oil particles were seeded uniformly within the region enveloping the thick and thin oil slicks with no distinction. The shapefiles were used here for both initialization of the oil drift simulations and for verification of results.

**2.2 Metocean forcing**

In the cases presented here, the ocean circulation fields come from a data-assimilative, high-resolution (1/50º, 1.8 km) configuration of the Hybrid Coordinate Ocean Model (HYCOM - www.hycom.org) in the Gulf of Mexico (GoM), developed by the authors. This configuration, which we refer to as GoM-HYCOM 1/50º, uses daily river forcing and data assimilation (GoM-HYCOM 1/50º). The HYCOM model has a flexible, hybrid vertical coordinate system, in which the distribution of vertical layers is optimized: they are isopycnal in stratified water columns, sigma terrain-following in regions with sharp topography, and isobaric in the mixed layer and very shallow areas (Bleck, 2002). More information about the HYCOM model is available in the model user's manual www.hycom.org and the references therein. The GoM-HYCOM 1/50º covers the entire GoM and uses 32 vertical levels. The model configuration is similar to the one used by Le Hénaff and Kourafalou (2016), with the realistic river forcing parameterization developed by Schiller and Kourafalou (2010). The river discharge data were obtained from the Army Corps of Engineers and the U.S. Geological Survey (USGS - www.usgs.gov). The model is initialized in October 2009 with fields from the operational Global HYCOM (GLB-HYCOM) simulation run at the Naval Research Laboratory at the Stennis Space Center (GLB-HYCOM expt_90.8, www.hycom.org), and it is nested at the open boundaries with model fields from the same simulation. The atmospheric forcing is based on the 3-hourly winds, thermal forcing and precipitation forecast

fields from the European Centre for Medium-Range Weather Forecasts (ECMWF www.ecmwf.int), with spatial resolution of 0.125º (see below). The model assimilates satellite observations of Sea Surface Temperature and Sea Surface Height, and in situ observations of temperature and salinity from buoys, cruises, surface drifters, Argo floats and XBT casts. More details about the model configuration can be found in Le Hénaff and Kourafalou (2016) and Androulidakis et al. (2018a). For the

5 present study, two simulations were performed: one with the attributes mentioned above, called *Reference* simulation, and one called *No river*, in which the salinity fronts have been removed by shutting off the river discharge, setting precipitation to zero, and turning off the assimilation of salinity profiles. All the other forcing conditions (e.g. meteorological, boundary) remained the same between the two experiments in order to investigate the impact of an individual forcing mechanism (here the Mississippi buoyant discharge and the related density fronts) on the circulation features and furthermore on the oil spill extensions

10 during the DwH period. The outputs from both simulations are available at the Gulf of Mexico Research Initiative Information and Data Cooperative (GRIIDC - data.gulfresearchinitiative.org/).

[revised manuscript text omitted]

Following Li et al. (2017b), the volume (V) droplet size spectrum is described by the median droplet diameter, $D_{50}^V$, as

$$D_{50}^V = d_o r \left(1 + 10\text{Oh}\right)^p \cdot \text{We}^q \tag{1}$$

20 with the empirical coefficient $r = 1.791$ and the exponents $p = 0.460$ and $q = -0.518$. The PDF for the droplet size distribution follows a log-normal distribution around the medium diameter with a logarithmic base-10 standard deviation of $s = 0.38$ (Eq. 16 in Röhrs et al. (2018)).

The Weber number, We, is a dimensionless number describing the relative importance of inertial forces and oil-water interfacial tension. It is a function of the sea water density, $\rho_w$, the significant wave height, $H_s$, and the oil-water interfacial tension,

25 $\sigma_{o-w}$, and is given by

$$\text{We} = \frac{\rho_w g H_s d_o}{\sigma_{o-w}}, \tag{2}$$

where $g$ is the acceleration of gravity and $d_o = 4\sqrt{\frac{\sigma_{o-w}}{g(\rho_w - \rho_o)}}$ is the Rayleigh-Taylor instability maximum diameter.

The Ohnesorge number, Oh, is a dimensionless number describing the ratio of viscous forces to inertial and surface tension forces. It is a function of the dynamic oil viscosity, $\mu_o$, oil density, $\rho_o$, and oil-water interfacial tension:

$$\text{Oh} = \frac{\mu_o}{\sqrt{(\rho_o \sigma_{o-w} d_o)}}. \tag{3}$$

30

The volume size distribution, following (Delvigne and Sweeney, 1988), is given by

$$V(d) = d^{-0.6} \text{for} d_{min} < d < d_{max} \tag{4}$$

where d is the droplet diameter. Minumum and maximum droplet radii are set to 10e-6 and 10e-3, respectively. The exponent of $-0.7$ in the volume size distribution corresponds to an exponent in the number size distribution of 2.3 (Tkalich and Chan, 2002).

Droplet sizes are assigned to oil particles each time a particle is submerged by breaking waves, following the wave entrainment algorithm of (Li et al., 2017c). The implementation of this algorithm in OpenOil is described if full detail in (Röhrs et al., 2018). The droplet sizes for individual particles are drawn from a random distribution according to the chosen size distribution. The size distributions represent conditions for a stochastic wave entrainment event, representing equilibrium conditions during a model time step. It is noted that the overall size distribution of all submerged oil in the simulation is further subject to changes, as weather conditions and the oil's emulsification rate change and oil droplets of various sizes are subject to various resurfacing time scales. Resurfaced particles are considered to be part of a surface slick, and are assigned a new droplet size once they are re-entrained. Oil elements at the sea surface (slick) are not considered to have a radius.

**Droplet size distribution during deep blowouts**

For oil elements released at the seafloor (wellhead), a simplistic and pragmatic approach of prescribing random radii in the range $0.5\,mm$ to $5\,mm$ was used, as suggested by Johansen (2000).

**Horizontal transport**

With regard to horizontal drift, three processes are considered: Any element, whether submerged or at the surface, drifts along with the ocean current. Elements are further subject to Stokes drift corresponding to their actual depth. Surface Stokes drift is normally obtained from a wave model, and its decline with depth is calculated as described in Breivik et al. (2016). Oil elements at the ocean surface are additionally moved with a factor of 2% of the wind. Together with the Stokes drift (typically 1.5% of the wind at the surface), this sums up to the commonly found empirical value of 3.5% of the wind speed (Schwartzberg, 1971). The magnitude of the wind drift factor is discussed in Jones et al. (2016) who stated that a 2% wind drift factor was required in OpenOil to reproduce their observations of a SOP in the North sea. In essence, this is believed to be a compensation factor for the inability of any ocean model to represent the strong shear current in the upper few centimeters/decimeters of the ocean, and not surface oil actually moving relative to the water.

The three horizontal drift components may lead to a very strong gradient of drift magnitude and direction in the upper few meters of the ocean. For this reason, it is also of critical importance to have a good description of the vertical oil transport processes.

**Vertical transport**

Oil elements at the surface, regarded as being in the state of an oil slick, may be entrained into the ocean by breaking waves. The entrainment of oil droplets depends on both the wind and wave (breaking) conditions, but also on the oil properties, such as viscosity, density and oil-water inter facial tension. The buoyancy of droplets is calculated according to empirical relationships and the Stokes law following Tkalich and Chan (2002), dependent on ocean stratification based on temperature and salinity from the ocean model, and the viscosities and densities of oil and water.

In addition to the wave induced entrainment, the oil elements are also subject to vertical turbulence throughout the water column, described using a random-walk scheme based on the turbulent eddy diffusivity from the ocean model.

**Weathering**

In order to calculate weathering of the oil, OpenOil interfaces with the open source ADIOS oil library, developed by NOAA (github.com/NOAA-ORR-ERD/OilLibrary) and (Lehr et al., 2002). In addition to state-of-the-art parametrization of weathering processes such as evaporation, emulsification and dispersion, this software contains a database of measured properties of almost 1000 oil types from around the world. As oils from different sources or wells have vastly different properties, such a database is of vital importance for accurate results. The ADIOS oil library is also used by the NOAA oil drift model github.com/NOAA-ORR-ERD/PyGnome.

The weathering algorithms describes evaporation and emulsification rate of oil, i.e. the water content. The emulsification and evaporation greatly affect oil density, viscosity and oil-water interfacial tension, and thereby the droplet size distribution through Eqs. 1-3. OpenOil take into consideration weathering processes that are dominating in the initial oil spill period of 2-3 days. Long-term weathering processes such as sedimentation and microbial degradation are not considered in this study.

**3   Results**

A first set of simulations are carried out to investigate the effect of oil droplet size distribution. A second set focuses on the effect of river induced fronts. According to Crone and Tolstoy (2010), the average flow rate from the oil well between 22 April and 3 June 2010 was estimated to 0.1 m$^3$ sec$^{-1}$, assuming a liquid oil fraction of 0.4. Gas and highly volatile compounds are not considered here. After the riser was removed and until the leak was sealed on 15 July, the flow rate increased to 0.12 m$^3$ sec$^{-1}$, corresponding to 10,368 m$^3$ day$^{-1}$. An assumption was made on of how much oil was present at the sea surface at the initialization of the simulation. The residence of oil at the sea surface depends heavily on oil properties as well as environmental conditions such as temperature of ocean and air, wind and waves, as described above. According to our mass balance calculations for the DWH spill, using the Light Louisiana Sweet oil type from the NOAA oil library and environmental conditions as described above, it seems reasonable to assume that 80% of the oil mass is removed from the surface after 10 days. This is within the range in our simulations that is typically 60 to 95% (see examples of mass balance plots further down). While Boufadel et al. (2014) assumed a constant removal rate of surface oil 20% per day, the removal rate in the present

simulation and in reality will vary with wind and wave conditions. The simulations for May 2010 are initialized by seeding 48730 particles in a polygon obtained from NOAA shapefiles. Each particle represents initially 1 m$^3$ oil. A continuous point source at the sea floor seeds an additional 8460 particles (8460 m$^{-3}$) per day during the simulation. After June 3rd these numbers are increased by 20% to 10368 m$^3$ day$^{-1}$. The oil elements released at the surface are assigned droplet radii at each entrainment

5    incident, according to the parameterisation of size distributions from respectively DS88 or Li17, see Röhrs et al. (2018) for details. Oil elements at the sea surface (slick) are not considered to have a radius.

Around 20-25 May 2010 there was a significant outflow of the Mississippi River (Fig. 2) and part of the SOP was entrained along the LC resulting in a formation popularly referred to as the "tiger tail" (Fig. 3). The OpenOil simulation shown on top in Fig. 3 is carried out using the classical DS88 oil droplet size distribution (Delvigne and Sweeney, 1988). Fig. 3 lower panel

10   shows the results from repeating this simulation using the new Li17 formulation. In Fig. 4 the mass balance during seven days for the Li17 simulation is shown. There is virtually no difference between DS88 and Li17 and only Li17 is shown here. Both formulations result in about the same fraction of oil at the surface (about 50%) after 7 days, with moderate wind speeds of up to 7.2ms$^{-1}$. The light compounds evaporate fast after release hence the more heavy compounds are tracked here ("dead oil"). Patches of thick oil (where the particles retain nearly 100% of their mass) are visible over a larger area (Fig. 3). Larger oil

15   droplets will rise faster to the surface (North et al., 2011; Röhrs et al., 2018), and DS88 provides a much higher fraction of large droplet after one hour compared to Li17 (Fig. 5, upper panels). However, it turns out that the DS88 and Li17 provide similar volume distributions after a 24 hr test simulation (Fig. 5, lower panels). Still, the peak in the distribution is at around 100 $\mu m$ for both formulations, and rapid rising to the surface can be expected according to North et al. (2011). Due to the small difference between DS88 and Li17, the DS88 simulation results in just marginally more oil stranded after seven days (13.2

20   vs 12.8 %), particularly west of the Mississippi Delta (Figs. 3). A higher fraction of oil at the surface provides more efficient transport by wind and waves towards the shore and larger likelihood of stranding. For both simulations, the oil particles quickly loose 40-50% of their mass, mostly through evaporation.

Fig. 6 shows the geographical distribution of particle diameters at the surface. It is apparent that smaller particles are present outside the MAFLA shelf, probably because the oil particles have been subject to more wind and hence wave action and natural

25   dispersion in the last 24 hours (Fig. 7).

[revised manuscript text omitted]

To the best of our knowledge, this is the first time the importance of the effect of river fronts on oil slick transport in the Gulf of Mexico has been demonstrated using high resolution forcing and a fully fledged oil drift model.

TEXT

*Code availability.* github.com/OpenDrift/opendrift

*Data availability.* TEXT

*Code and data availability.* TEXT

*Sample availability.* TEXT

*Video supplement.* TEXT

**Appendix A**

**A1**

5 *Author contributions.* TEXT

*Competing interests.* TEXT

*Disclaimer.* TEXT

*Acknowledgements.* This research was made possible by a grant from The Gulf of Mexico Research Initiative (award "Influence of river induced fronts on hydrocarbon transport," GOMA 23160700). Atmospheric and wave data were kindly provided by the European Center 10 for Medium-Range Weather Forecasts (ECMWF). M. Le Hénaff acknowledges partial support from the Physical Oceanography Division at NOAA's Atlantic Oceanographic and Meteorological Laboratory, AOML. The outputs from the HYCOM simulations are available at the Gulf of Mexico Research Initiative Information and Data Cooperative (GRIIDC - doi:10.7266/N7NG4NPC).

[revised manuscript text omitted]

---

## Author Comment (AC3) · 20 Feb 2019

[12pt]article   We would like to thank this reviewer for taking the time to provide very detailed and specific comments to our manuscript. Below are our responses with the reviewers comments in bold letters.

**General Comments This paper presents simulations of oil transport during the Deepwater Horizon (DWH) accident using a Lagrangian Particle Tracking model with oil fate processes (OpenOil) coupled to an ocean circulation, wave, and wind model (GoM-HYCOM, WAM, and ECMWF), run under different freshwater discharge scenarios through the Mississippi delta system. While such a study**

**is important and in general the modeling systems used are adequate for the task, the present manuscript lacks the information needed to critically assess the modeling results.**

**Very broadly, the article lacks the clarity needed to understand both what the authors have done and how their models work. In Section 2, where the modeling system is described, references are missing to external model products (e.g., ECMWF results) and the text is unclear whether standard model products are used or whether custom simulations were conducted for this work. Where custom simulations are done, there needs to be better referencing to the base model and citations to any work validating the custom model. These problems should be easily resolved by re-writing this section to become much more clear. Indeed, poor writing quality is a problem throughout this manuscript, and most of the detailed comments below stem from vague or difficult to understand sentences throughout the text.**

We use ECMWF standard forecast products for atmosphere and wave. This is explained in the "Metocean forcing" section we where refer to ECMWF publications and web site where we have obtained the products. Temporal and spatial resolution is also given in this section. We have to tried to state this even clearer and used the term "forecast products" instead of "forecasts".

**The greatest technical deficiency stems from the description of OpenOil and its usage. We have a long paragraph describing OpenOil with several references. We have now provided more details on how the oil droplet size distribution is prescribed in several new subsections.**

The detailed comments below give several, specific issues related to the handling of this model in the text. In summary, the following elements need to be improved before the model results can be critically evaluated:

**The authors need to better distinguish surface and sub-surface sources of oil**

and describe the chemical properties of the oil as initialized in the model. I as-
sume the authors use the dead-oil representation of LSC from ADIOS for both
surface and subsurface sources. This is ok, but they need to clearly state this
and explain what the flow rate at the subsurface source means (it neglects all
gas). Yes we use the dead-oil representation of LSC. This is now clearly stated. Here,
we track only surface oil particles. This is also now clearly stated.

**The authors need to provide the equations for DS88 and Li17 since they are
single-line, empirical formulas. This will greatly improve understanding for the
reader.** We agree. This is included now. Several new subsubsections are now
included describing the droplet size distribution calculation.

**The authors need to explain how OpenOil uses DS88 and Li17 to predict a time-
evolving droplet size distribution. Both of these equations are empirical formu-
las for the steady-state size distribution of submerged oil under various sur-
face wave conditions. It is not obvious how these models produce time-variable
droplet size distributions.** The time-dependency is a result of longer time integra-
tion in the full oil spill model. The droplet size distribution from DS88 and Li17 are here
assumed to apply after stochastic wave breaking events in the model, thus correspond-
ing to equilibrium spectra from lab tank experiments. We argue that during the further
evolution in the full model, turbulence and buoyant resurfacing of particles continues
to affect the total droplet size distribution because larger droplets resurface faster that
small particles. Resurfaced particles are subject to subsequent wave breaking and
resurfacing. The total droplet size spectrum therefore changes on the course of hours
to days, which is not achieved in laboratory experiments. In the full model, oil proper-
ties also change due to further emulsification and and varying weather conditions. In
the present figure 5, we start out with a random distribution and use a constant wind
speed of 8 m/s over some time in order allow emulsification to evolve. The figure shows
that the two formulations use some time to reach steady-state.

**Only Li17 predicts oil droplet size for a jet (e.g., the subsurface source), and**

**technically, this is a different equation (different fit parameters) to the surface size distribution in that paper. What equation was used for subsurface oil? How was this done in the DS88 simulations?** How droplet sizes for the surface release are described in OpenOil, is documented in a very recent paper in OS (Röhrs et al. 2018). For the bottom release, we have unfortunately overseen to describe how particle distributions are initialized, and would like to thank the reviewer for spotting this shortcoming. We have now added the following paragraph to Section 3 to clarify how droplet sizes are given: "The oil elements released at the surface are assigned random droplet radii at each entrainment incident, according to the parameterisation of size distributions from respectively DS88 or Li17, see Röhrs et al. (2018) for details. For oil elements released at the seafloor (wallhead), we use a simplistic and pragmatic approach of prescribing random radii in the range 0.5mm to 5mm, as suggested by Johansen et al. (2000). Oil elements at the sea surface (slick) are not considered to have a radius."

**The fate processes considered in the model are applicable to surface oil. They can also be used for subsurface oil when dead oil is introduced into the model (which I believe the authors have done). However, dissolution, which is neglected in the present study, was a significant fate process during DWH and occurred very quickly. The authors are only justified to ignore it because they track only the low-solubility components of the oil.** We agree with the reviewer. We use the flow rate of Crone and Tolstoy (2010) ignoring gas and highly volatile compounds. This is now clearly stated in the text.

**After these important issues are resolved, the results can be critically evaluated. Throughout the results, the authors should carefully connect droplet sizes to mech- anisms. For example, in point 25.) below the authors state that small droplets were subject to greater wind forcing. In what way? Perhaps this is true, but the mechanism is unstated. My reading of this does not make sense because I would assume smaller particles move away from the sea surface, where they**

**are less impacted by winds. But, another reading could be that smaller particles are generated by stronger winds. Yet, the manuscript does not really explain it either way.** We have now improved our discussion of the figures involving droplet diameter and wind speed (present Figs 6 and 7). What we meant was that these figures show that the droplets exposed to the highest wind speed have the smallest diameter, in accordance with theory. We do not mean that smaller droplets are subject to more wind drift.

**Overall, my conclusion is that this is an important study conducted by researchers who are clearly capable of handling the ocean circulation under different freshwater inflows. They should spend more time polishing the text and much more carefully explain the droplet size modeling, handle the complex thermodynamics of the oil and gas, and more carefully assess and describe the important fate mechanisms. The remainder of this review addresses specific elements of the manuscript text. 1. In the abstract, the authors describe the droplet size analysis and the key result (that the results are robust to the choice of algorithm). However, for the effects of river inflow, the abstract lacks the key result. It would be best to also summarize the main outcome of that work within the abstract.** We agree. This is included now.

**2. The introduction is a collection of quite detailed sentences that do not flow together well. While each sentence is well referenced, I wonder whether the normal reader can follow these statements. For instance, for the Taylor Energy study, the paper reads, "The drifters deployed during the experiment...efficiently described three major transport pathways;" yet, the text does not introduce or describe the drifters before this key result statement. It would be better to say earlier in the paragraph that the Taylor Energy study included deployment of surface drifters and explain whether they were all released at the source? all at the same time? on what dates or in which season? what layer of water is tracked? As is, the reader really needs to know more about this study than is stated to understand**

**the text. This is a specific comment that can be generalized to most of the statements in the introduction. This part of the paper could be significantly improved by more careful and tight writing.** The introduction has been rewritten with several more references, trying to be more concise and to the point.

**3. Page 2, Lines 21-25. This section is a little misleading. In the abstract and later in the paper, the reader will realize that oil droplets are initialized at the surface and at the seabed. The model does include the key fate factors for surface oil, but does not include the key fate factor for the subsurface oil (dissolution). This section should be revised to remove this ambiguity. As is, it sounds like the model includes the key fate factors for all oil droplets released in the model; this is only true for the surface released oil or the list of factors stated in this sentence is incomplete (it should also include dissolution).** We agree. The have now rewritten this paragraph and explained that dissolution is not included.

**4. Page 2, Line 33. Include a citation with a link to the ERMA website and the date the data were accessed / downloaded.** Not sure what is meant. This was already in the reference list, but the link is now given in the text.

**5. Page 3, Line 2. "We used the oil thickness classifications derived from the satel- lite analysis for our modeling study." How were these used? The next sentence says that particles were seeded without respect to thickness. At least give the reader some hint here how thickness was used and where in the manuscript this concept will be expanded.** Our apologies. This statement was included by mistake. We do not consider oil film thickness in this study.

**6. Page 3, Line 6. It seems that this sentence needs a citation to the source where the HYCOM model data, or at least numerical grid and initial conditions, were downloaded. The statements starting on Line 9 hint that the model simulations used in this paper may have been generated by the authors for this paper. Is that the case? If so, this should be explicitly stated. GoM-HYCOM is also an**

**operational model, so a reader may assume that standard HYCOM products were used. The last sentence seems to make this more clear (maybe the authors use their own output). I recommend revising this paragraph to make it even more clear that the authors performed custom simulations using this modeling system, state what elements are from standard GoM-HYCOM products, clearly explain the initial conditions, and explain whether data assimilation is turned on for any of the present simulations.**

We agree that that paragraph was not totally clear in the initial version. We did use model outputs from our own configuration of HYCOM over the Gulf of Mexico, which is data-assimilative. This is now stated more clearly in the revised version: "In the cases presented here, the ocean circulation fields come from a data-assimilative, high-resolution (1/50o, 1.8 km) configuration of the Hybrid Coordinate Ocean Model (HYCOM) in the Gulf of Mexico (GoM), developed by the authors (GoM-HYCOM 1/50)." We provide additional information about the initial conditions: "The model is initialized in October 2009 with fields from the operational Global HYCOM (GLB-HYCOM) simulation run at the Naval Research Laboratory at the Stennis Space Center (GLB-HYCOM expt_90.8, HYC) [. . .]." Finally, we now clearly indicate where data from both simulations using our Gulf of Mexico model configuration can be accessed on the GRIIDC data server: "The outputs from both simulations are available at the Gulf of Mexico Research Initiative Information and Data Cooperative (GRIIDC), doi: 10.7266/N7NG4NPC."

**7. Much of the text could be much better polished. A good example is the sentence starting on Page 3, Line 9, "The HYCOM model has..." In this sentence, the different layers are described. Use parallel grammatical structure: "Isopycnal layers in stratified water", "sigma terrain-following layers in sharp topography", then edit the next phrase to read "isobaric layers in the mixed layer and very shallow areas." Finally, the format of the citation needs to be within parentheses, e.g., (Bleck, 2002). While this does not impact the scientific merit of the paper,**

**these types of careless mistakes distract from the impact of the paper and make illustration of sloppy writing throughout the manuscript.** We have corrected the sentence following the reviewer's suggestion. It now reads: "The HYCOM model has a flexible, hybrid vertical coordinate system, in which the distribution of vertical layers is optimized: they are isopycnal in stratified water columns, sigma terrain-following in regions with sharp topography, and isobaric in the mixed layer and very shallow areas (Bleck, 2002). This unique utility in combination with the special treatment of freshwater inputs (Schiller and Kourafalou, 2010), also make HYCOM advantageous in areas with complicated topography, such as the GoM, and strong freshwater outflows such as the MR discharge, allowing the development of detailed process studies around the outflow regions, where plume dynamics are dominant (Androulidakis et al., 2015)." We have corrected the typo in the reference to Bleck.

**8. Inconsistent usage of ECM and ECMWF (Page 3, Lines 18 and 25). Also, provide a citation for where and when the ECMWF data were downloaded.** ECMWF is now used throughout, and the link is also provided in the text.

**9. Page 3, Line 28. Oil simulations were performed with a 3-hourly time step; yet, the text says that ECMWF forcing data have a 12 hourly time step. How was this data sub-sampled? Sample and hold? Linear interpolation? Other?** We used linear interpolation between time steps for waves. Atmospheric data are provided with 3 hour time step. This is now stated in the text.

**10. Page 4, Line 1. Provide a citation to the data source for the WAM data that were downloaded from ECMWF.** Reference to ECMWF was already given in the reference list and is now given in the text.

**11. Section 2.2. As one reads this entire section it is very difficult to separate model- ing work done by the present authors from modeling work conducted elsewhere and used by the present authors. Perhaps it would be helpful for each model product to start with a statement that either reads something like,**

"**Wave proper- ties were downloaded from WAM model results available from ECMWF (citation),**" **or** "**We simulated wave forcing using the WAM model.**" **Also, the acronym WAM is undefined.** We have rewritten according to the reviewers suggestion. The acronym WAM is now clearly stated in the metocean section and we cite ECMWF wind and wave data separately.

**12. Page 4, Line 18.** "**Droplet size spectra**" **should probably be** "**Droplet size distribu- tion**". **Reserve the term** "**spectrum**" **to refer to energy distributions. Page 4, Line 18ff. The Delvigne and Sweeney (1988) model must predict two parameters of the droplet size distribution: a characteristic droplet size and a variance parameter. The text appears to discuss the variance parameter, as given by a power-law relationship. I think this section would be much easier to understand if the equation is provided, e.g., P DF = ad b . Then, the authors can also discuss the proportionality constant (i.e., a) and the state variable of the power law would be defined (presently, it is undefined, and I assume it is droplet size, d).** We have now written "droplet size distribution" throughout. Much more details are provided on the oil droplet distribution including several equations.

**I am uncertain under what conditions Li17 would give d 50 =100** $\mu$**m. That is a very small size that would result from high mixing energy and low interfacial tension. Rather than state** "**depending on oil and environmental conditions**"; **instead state,** "**for an oil with X interfacial tension and Y viscosity, in a sea state of Z, ...**". **As is, this formulation is very misleading since it is difficult to get oil droplets this small without intervention (e.g., chemical dispersant application).** Please see comment above. Our results are for example in accordance with North et al (2011).

**15. Page 4, Line 34.** "**physical mechanism...is not obvious.**" **On the contrary, the physical mechanism for wind drift is well known and understood. Oil typically occupies a region very close to the water surface or on the surface (e.g., slicks). Numerical models of ocean circulation have large vertical layer thickness at the**

[Figure]

surface (about 50 cm or more) relative to the oil (about 1 mm to 10 cm). If the **numerical model completely resolved the surface boundary layer, there would be no added wind drift. Because the circulation model integrates the top 50 cm to 1 m, there needs to be an empirical wind drift added. 3value only because circulation models have very similar vertical layer thicknesses in their numerical discretization. If a model had a unique surface discretization, a different empirical factor would be needed.** We agree. What we meant is that the size of the wind drift factor can be discussed. More details from Jones et al (2016) and discussion are provided now.

**16. Top of Page 5. Unusual usage of a colon at the end of the first paragraph. Re- place with ""...the sum of oil entrainment by breaking waves and vertical turbulent diffusion." Also, "inter facial" should be "interfacial". Finally, the present gram- matical construction makes it seem as if the ADIOS mechanisms are part of the processes describing vertical transport, which they are not. Instead, ADIOS is a 0-D model that predicts oil fate.** We agree. This paragraph has been rewritten.

**17. Page 5, Line 18. Dissolution is by no means a "long-term weathering process." Dissolution is largely complete in the time span of minutes to hours and much less than one day. Dissolution is ignored in surface fate models because it is typically about 10 times less important than volatilization. For oil not in contact with the air/water interface (e.g., oil discharged at the seafloor), dissolution is the dominant fate process, and is the main fate mechanism for 100discharged from the DWH and for about 25spilled. This dissolution occurred as oil droplets and gas bubbles transported from the riser to the sea surface, a process that took about 4 to 12 hours duration (see papers by Ryerson in GRL and PNAS). Dissolution cannot be listed in this sentence as a long-term fate process, nor can it be completely ignored for DWH. The present authors may ignore dissolution if 1.) they assume it is fast and 2.) they track the insoluble components of**

**the oil.** We agree with the reviewer. As seen in the mass balance plot (now Fig. 4), most of the oil we track is at the surface. Hence the volatile compounds evaporate fast (as seen in Fig. 4), and we track the heavy compounds.

**18. Page 5, Line 23. The flow rate provided is the flow rate of crude oil (dead oil), neglecting the release of light compounds (e.g., gas). The sentence reads as if the 0.1 m 3 /s is the in situ flow rate of oil and liquid petroleum at the well head. The meaning of the flow rate needs to be very carefully defined. Since the model neglects dissolution, I believe this is the correct flow rate to use at the seafloor. I am just asking the authors to carefully define it—it is the flow rate of dead (black) oil, neglecting gases and highly-volatile compounds.** We follow Crone and Tolstoy (2010). They estimate a liquid oil fraction of 0.4 and get a flow rate of 0.1 m-3 sec. We now state clearly that gas and highly volatile compounds are not included here.

**19. Page 5, Line 24. "Although..." I think you mean "Because..."** We agree. Corrected.

**20. Page 5, Line 28. Rather than submit a new estimate for the removal rate of oil from the surface, can the authors not cite another study? Does the Oil Budget Calculator (OBC) contain this estimate? Many subsequent papers have confirmed the estimates in the OBC.** We think it is most consistent to use a removal rate in line with our model study. It will vary from day to day with wind and wave conditions. This removal rate is only required for estimating the amount of oil present at the surface when the model is initiated (from shape files). Most other studies have used constant removal rates as far as we can see.

**21. Page 5, Line 31. Define "super-particles". I know what they are, but they are not defined as far as I can tell in the paper.** We now just write particles through out the manus.

**22. Page 5, Line 32. What is the droplet size used for the particles seeded at the**

[Figure]

**seafloor? Only the Li17 equation can predict droplet sizes from a jet discharge, but technically, the Li17 equation is two equations: one for a jet and one for surface oil. Did the authors use both? Later, results are compared between Li17 and DS88. For the particles released at the seafloor, this comparison should be meaningless as DS88 does not provide estimates for a jet. As such, this comparison becomes difficult to assess.** This is now explained. For oil elements released at the seafloor (wallhead), a simplistic and pragmatic approach of prescribing random radii in the range 0.5mm to 5mm was used, as suggested by Johansen (2000).

**23. Page 6, Line 5. One would not expect much difference among the models since they are both calibrated to the same experimental data. Perhaps the authors can explain a hypothesis why these models might be expected to give different re- sults? For instance, though both may predict the same mean droplet sizes, they use different distributions. If the fate of oil in the tails of the distributions is differ- ent, then maybe one would expect differences. But, since Li17 is calibrated to the same data as DS88, I would not expect a difference a priori. And, if differences are only seen in the tails of the distributions, one should consider whether this involves much mass.** A long section explaining the differences between the formulations is now provided.

**24. Page 6, Line 8-12. Both the DS88 and Li17 models are steady-state. I cannot understand how they could give different results after 1 hour to their results after 24 hours. The only way their results can change is if the ocean forcing changes, which may be happening, or if they are used iteratively with changing surface oil conditions. However, the text completely obscures this fact and implies that the models are unsteady, producing time-variable results. Li17 and DS88 are not and do not. This must have to do with the way OpenOil uses these in a real, time-evolving simulation. In that case, more details of OpenOil are needed, and because this is a critical conclusion of the paper, a citation to OpenOil would not be adequate: the key mechanisms / methods used to generate time-variable**

[Figure]

**droplet size distributions from steady-state equations must be explained.** The time-dependency is a result of longer time integration in the full oil spill model. The droplet size distribution from DS88 and Li17 are here assumed to apply after stochastic wave breaking events in the model, thus corresponding to equilibrium spectra from lab tank experiments. We argue that during the further evolution in the full model, turbulence and buoyant resurfacing of particles continues to affect the total droplet size distribution because larger droplets resurface faster that small particles. Resurfaced particles are subject to subsequent wave breaking and resurfacing. The total droplet size spectrum therefore changes on the course of hours to days, which is not achieved in laboratory experiments. In the full model, oil properties also change due to further emulsification and and varying weather conditions. In the present figure 5, we start out with a random distribution and use a constant wind speed of 8 m/s over some time in order allow emulsification to evolve. The figure shows that the two formulations use some time to reach steady-state.

**25. Page 6, Line 15. Small particles will be easily entrained into the ocean interior and away from the surface. The text here says the small particles have probably Discussion paper"been subjected to more wind action in the last 12 hours." This sentence needs explanation. What affect of the wind? Do the authors mean the particles are smaller because the wind generated larger waves, yielding more energy, resulting in smaller particles? It does not sound like it. The paragraph reads to me that the authors are claiming the smaller particles experience greater wind drift. I would expect larger particles to experience greater wind drift as they will be closer to the surface. Please edit for clarity.** This has now been rewritten: ...MAFLA shelf, probably because the oil particles have been subject to more wind and hence wave action and natural dispersion in the last 24 hours

**26. Page 6, Line 25ff. These results are meaningful in so far as the oil droplet sizes are understood. The above comments need significant clarification for the oil droplet sizes before this section can be rigorously evaluated.** We agree. Much

more detail is now provided

**27. Page 8, Line 5. The conclusion is that the droplet size distribution has a signifi- cant effect on horizontal and vertical distributions and a wind speed range of 5-7 m/s is noted. This appears contrary to the results, which show two different size distribution models yield very similar results. Perhaps the authors mean that the fate of the oil is linked to the size distribution and they are not claiming the two models give different results. While I also agree size distribu- tion is critical, this paper does not show that result; instead, it shows that two reasonable models give similar results. To conclude that results would be much different with a differ- ent size distribution, additional simulations with a different size distribution would be needed.** We agree that this statement is misleading and have rewritten accordingly stating that "Our results indicate that the two different formu- lations for oil droplet size distribution give similar results for both vertical and horizontal distribution of the oil".

**28. I also note that this paper ignores all of the prior work using far-field La- grangian Particle Tracking models by Elizabeth North's and Claire Paris's re- search groups. While North generally uses SABGOM circulation forcing, Paris also uses GoM-HYCOM. This work should be summarized and critically reviewed in the Introduction. Key results of this work that confirm or deny similar con- clusions in North's or Paris's papers should also be highlighted throughout the Results and Discussion.** We thank the reviewer for pointing this out. The papers by North, Paris, Barker and others are now properly cited in the introduction, results and discussion

[Figure]

**Supplement:**

**Revisiting the DeepWater Horizon spill: High resolution model simulations of effects of oil droplet size distribution and river fronts**

Lars R Hole[1], Knut-Frode Dagestad[1], Johannes Röhrs[1], Cecilie Wettre[1], Vassiliki H. Kourafalou[2], Ioannis Androulidakis[2], Matthieu Le Hénaff[3,4], Heesook Kang[2], and Oscar Garcia-Pineda[5]

[1]Norwegian Meteorological Institute, Allegt. 70, 5007 Bergen, Norway
[2]University of Miami, Rosenstiel School of Marine and Atmospheric Science, Miami, FL, USA
[3]University of Miami, Cooperative Institute for Marine and Atmospheric Studies, Miami, FL, USA
[4]NOAA Atlantic Oceanographic and Meteorological Laboratory, Miami, FL, USA
[5]WaterMapping, Gulf Breeze, FL, USA

**Correspondence:** Lars Robert Hole (lrh@met.no)

**Abstract.** An open source ocean trajectory framework, OpenDrift, is used to simulate the 2010 DeepWater Horizon oil spill. Metocean forcing data are taken from the GoM-HYCOM 1/50º ocean model with realistic river input and ECMWF global forecast products of wind and wave parameters with 1/8º resolution. OpenDrift includes the integrated oil drift module OpenOil, which includes a number of relevant processes, such as emulsification, wave entrainment, and droplet formation. This takes account of the actual oil type/properties, using the ADIOS oil weathering database of NOAA. The effect of using a newly developed parameterization for oil droplet size distribution is studied, compared to a traditional algorithm. Although the algorithms provide different distributions for a single wave breaking event, it is found that the net difference after long time simulations is negligible, indicating that the outcome is robust regarding the choice of parameterization. In both cases, the size of the droplets controls how much oil is present at the surface and hence are subject to wind and Stokes drift. The oil droplet size is also relevant for the biological impact. Next, the effect of removing river outflow in the ocean model is investigated in order to showcase effects of river induced fronts on oil spreading. A consistent effect on the amount and location of stranded oil is found, and considerable impact of river induced fronts is seen on the location of the surface oil patch. During a case with large river outflow (May 20-27, 2010), the total amount of stranded oil is reduced by about 50% in the simulation with no river input.

*Copyright statement.* TEXT

**1 Introduction**

The presence of both shelf and open sea dynamics makes the Northern Gulf of Mexico (NGoM) a topographically and dynamically complex and interesting study area, in the presence of intense oil exploration. Interactions of the Mississippi River

(MR) plume and the Loop Current (LC) system were found to be important for the transport and fate of oil during the 2010 DeepWater Horizon (DWH) incident (Kourafalou and Androulidakis, 2013; Le Hénaff et al., 2012).

According to the U.S. Energy Information Administration, more than 45% of total U.S. petroleum refining capacity and 51% of total natural gas processing plant capacity is located along the Gulf coast www.eia.gov/special/gulf_of_mexico/. Oil leaks and accidents, such as the explosion on the DWH platform in 2010 (at 28.737ºN, 88.366ºW), have released significant quantities of hydrocarbons (Crone and Tolstoy (2010); McNutt et al. (2011)) in the sensitive marine environment around the MR Delta, and over the LouisianA TEXas (LATEX) and Mississippi Alabama FLoridA (MAFLA) shelves (Kourafalou and Androulidakis, 2013) (Fig. 1).

Many studies have dealt with simulation of the DWH spill (North et al., 2011; Mariano et al., 2011; MacFadyen et al., 2011; Barker, 2011; Le Hénaff et al., 2012; Paris et al., 2012; Kourafalou and Androulidakis, 2013), with focus on both subsurface (Paris et al., 2012) and surface (Le Hénaff et al., 2012) transport. North et al. (2011) used a plume model to predict a stratification-dominated near field, in which small oil droplets detrained from the central plume containing faster rising large oil droplets and gas bubbles and became trapped by density stratification. They showed that simulated droplets with diameters between 10 and 50 $\mu m$ formed a distinct subsurface plume, which was transported horizontally and remained in the subsurface for >1 month. In contrast, droplets with diameters >90 $\mu m$ rose rapidly to the surface. Le Hénaff et al. (2012) focused on oil transport on the water surface and found that the wind played a major role in advecting the oil to the northern GOM. Barker (2011) conducted Monte Carlo simulations consisting of 500 individual oil trajectory scenarios using historical data of water currents and winds. The results by Barker (2011) indicated that, in approximately 75% of the scenarios, oil would be transported out of the GOM by the Loop Current. This means that the actual trajectory of oil from the DWH falls in the 25% of scenarios.

Androulidakis et al. (2018b) carried out a field experiment deploying surface drifters at different times near the Taylor Energy Site which is located in the vicinity of the MR outflow region over the NGoM and near the DWH site (approximately at 28.938ºN, 88.978ºW, Fig. 1). This multi-platform observational experiment was conducted in April 2017 to investigate the main transport pathways from the Taylor Site and toward the NGoM continental shelves and offshore, toward the Gulf interior. Results indicated that the surface transport was determined by the MR plume extension over the Taylor Energy Site and the river induced fronts in combination with local circulation, prevailing winds and broader regional dynamics (LC system). The drifters deployed during the field experiment in tandem with satellite data, drone imagery, wind measurements, and marine radar derived currents and images described three major transport pathways, in agreement with the three major circulation patterns of the MR plume (Schiller and Kourafalou, 2010; Androulidakis et al., 2018a; Schiller et al., 2011; Schiller and Kourafalou, 2014).

The drifters deployed by Androulidakis et al. (2018b) followed the two prevailing coastal currents associated with MR plume dynamics (downstream/upstream moving westward/eastward of the Mississippi Delta) and an offshore pathway under the influence of basin-wide circulation. Near the Taylor site, the existence of multiple river fronts influence the fate of oiled waters, preventing the transport of hydrocarbon toward the delta like a natural boom barrier, trapping and directing the oil either westward or eastward in agreement with Kourafalou and Androulidakis (2013), who showed a similar interaction during

the DWH accident. *In situ* thermohaline measurements around the Taylor Energy Site and across the river front showed that the MR plume near the Taylor Site was 5m to 10m deep, while the clearer ocean water column was characterized by a 40 m upper-ocean homogeneous layer, mainly controlled by temperature.

In this study, an open source Lagrangian oil drift model, OpenOil, has been used to simulate the DWH oil spill evolution. OpenOil takes into account major factors that influence the short term drift of an surface oil slick such as metocean forcing (including Stokes drift), emulsification, evaporation and vertical entrainment and mixing. Dissolution, which is important for subsurface oil spills, is not yet implemented. Simulations are initiated from satellite observations and a point source at the sea floor. The effect of using two different oil droplet size distribution on the horizontal drift and vertical mixing is discussed. Next, the study showcases how NGoM oil pathways are influenced by river plume circulation and river induced fronts. It is also investigated whether the use of realistic daily river discharge has a significant effect on the simulated location of the Surface Oil Patch (SOP) and stranding of oil.

**2 Methods and data**

**2.1 Shapefiles of surface oil patch**

Shape files derived from satellite analysis of the DWH SOP can be accessed through the NOAA-ERMA website erma.noaa. gov/gulfofmexico. In the present study, oil particles were seeded uniformly within the region enveloping the thick and thin oil slicks with no distinction. The shapefiles were used here for both initialization of the oil drift simulations and for verification of results.

**2.2 Metocean forcing**

In the cases presented here, the ocean circulation fields come from a data-assimilative, high-resolution (1/50º, 1.8 km) configuration of the Hybrid Coordinate Ocean Model (HYCOM - www.hycom.org) in the Gulf of Mexico (GoM), developed by the authors. This configuration, which we refer to as GoM-HYCOM 1/50º, uses daily river forcing and data assimilation (GoM-HYCOM 1/50º). The HYCOM model has a flexible, hybrid vertical coordinate system, in which the distribution of vertical layers is optimized: they are isopycnal in stratified water columns, sigma terrain-following in regions with sharp topography, and isobaric in the mixed layer and very shallow areas (Bleck, 2002). More information about the HYCOM model is available in the model user's manual www.hycom.org and the references therein. The GoM-HYCOM 1/50º covers the entire GoM and uses 32 vertical levels. The model configuration is similar to the one used by Le Hénaff and Kourafalou (2016), with the realistic river forcing parameterization developed by Schiller and Kourafalou (2010). The river discharge data were obtained from the Army Corps of Engineers and the U.S. Geological Survey (USGS - www.usgs.gov). The model is initialized in October 2009 with fields from the operational Global HYCOM (GLB-HYCOM) simulation run at the Naval Research Laboratory at the Stennis Space Center (GLB-HYCOM expt_90.8, www.hycom.org), and it is nested at the open boundaries with model fields from the same simulation. The atmospheric forcing is based on the 3-hourly winds, thermal forcing and precipitation forecast

fields from the European Centre for Medium-Range Weather Forecasts (ECMWF www.ecmwf.int), with spatial resolution of 0.125º (see below). The model assimilates satellite observations of Sea Surface Temperature and Sea Surface Height, and in situ observations of temperature and salinity from buoys, cruises, surface drifters, Argo floats and XBT casts. More details about the model configuration can be found in Le Hénaff and Kourafalou (2016) and Androulidakis et al. (2018a). For the

5 present study, two simulations were performed: one with the attributes mentioned above, called *Reference* simulation, and one called *No river*, in which the salinity fronts have been removed by shutting off the river discharge, setting precipitation to zero, and turning off the assimilation of salinity profiles. All the other forcing conditions (e.g. meteorological, boundary) remained the same between the two experiments in order to investigate the impact of an individual forcing mechanism (here the Mississippi buoyant discharge and the related density fronts) on the circulation features and furthermore on the oil spill extensions

10 during the DwH period. The outputs from both simulations are available at the Gulf of Mexico Research Initiative Information and Data Cooperative (GRIIDC - data.gulfresearchinitiative.org/).

[revised manuscript text omitted]

Following Li et al. (2017b), the volume (V) droplet size spectrum is described by the median droplet diameter, $D_{50}^V$, as

$$D_{50}^V = d_o r \left(1 + 10\text{Oh}\right)^p \cdot \text{We}^q \tag{1}$$

20 with the empirical coefficient $r = 1.791$ and the exponents $p = 0.460$ and $q = -0.518$. The PDF for the droplet size distribution follows a log-normal distribution around the medium diameter with a logarithmic base-10 standard deviation of $s = 0.38$ (Eq. 16 in Röhrs et al. (2018)).

The Weber number, We, is a dimensionless number describing the relative importance of inertial forces and oil-water interfacial tension. It is a function of the sea water density, $\rho_w$, the significant wave height, $H_s$, and the oil-water interfacial tension,

25 $\sigma_{o-w}$, and is given by

$$\text{We} = \frac{\rho_w g H_s d_o}{\sigma_{o-w}}, \tag{2}$$

where $g$ is the acceleration of gravity and $d_o = 4\sqrt{\frac{\sigma_{o-w}}{g(\rho_w - \rho_o)}}$ is the Rayleigh-Taylor instability maximum diameter.

The Ohnesorge number, Oh, is a dimensionless number describing the ratio of viscous forces to inertial and surface tension forces. It is a function of the dynamic oil viscosity, $\mu_o$, oil density, $\rho_o$, and oil-water interfacial tension:

$$\text{Oh} = \frac{\mu_o}{\sqrt{(\rho_o \sigma_{o-w} d_o)}}. \tag{3}$$

30

The volume size distribution, following (Delvigne and Sweeney, 1988), is given by

$$V(d) = d^{-0.6} \text{for} d_{min} < d < d_{max} \tag{4}$$

where d is the droplet diameter. Minumum and maximum droplet radii are set to 10e-6 and 10e-3, respectively. The exponent of $-0.7$ in the volume size distribution corresponds to an exponent in the number size distribution of 2.3 (Tkalich and Chan, 2002).

Droplet sizes are assigned to oil particles each time a particle is submerged by breaking waves, following the wave entrainment algorithm of (Li et al., 2017c). The implementation of this algorithm in OpenOil is described if full detail in (Röhrs et al., 2018). The droplet sizes for individual particles are drawn from a random distribution according to the chosen size distribution. The size distributions represent conditions for a stochastic wave entrainment event, representing equilibrium conditions during a model time step. It is noted that the overall size distribution of all submerged oil in the simulation is further subject to changes, as weather conditions and the oil's emulsification rate change and oil droplets of various sizes are subject to various resurfacing time scales. Resurfaced particles are considered to be part of a surface slick, and are assigned a new droplet size once they are re-entrained. Oil elements at the sea surface (slick) are not considered to have a radius.

**Droplet size distribution during deep blowouts**

For oil elements released at the seafloor (wellhead), a simplistic and pragmatic approach of prescribing random radii in the range $0.5\,mm$ to $5\,mm$ was used, as suggested by Johansen (2000).

**Horizontal transport**

With regard to horizontal drift, three processes are considered: Any element, whether submerged or at the surface, drifts along with the ocean current. Elements are further subject to Stokes drift corresponding to their actual depth. Surface Stokes drift is normally obtained from a wave model, and its decline with depth is calculated as described in Breivik et al. (2016). Oil elements at the ocean surface are additionally moved with a factor of 2% of the wind. Together with the Stokes drift (typically 1.5% of the wind at the surface), this sums up to the commonly found empirical value of 3.5% of the wind speed (Schwartzberg, 1971). The magnitude of the wind drift factor is discussed in Jones et al. (2016) who stated that a 2% wind drift factor was required in OpenOil to reproduce their observations of a SOP in the North sea. In essence, this is believed to be a compensation factor for the inability of any ocean model to represent the strong shear current in the upper few centimeters/decimeters of the ocean, and not surface oil actually moving relative to the water.

The three horizontal drift components may lead to a very strong gradient of drift magnitude and direction in the upper few meters of the ocean. For this reason, it is also of critical importance to have a good description of the vertical oil transport processes.

**Vertical transport**

Oil elements at the surface, regarded as being in the state of an oil slick, may be entrained into the ocean by breaking waves. The entrainment of oil droplets depends on both the wind and wave (breaking) conditions, but also on the oil properties, such as viscosity, density and oil-water inter facial tension. The buoyancy of droplets is calculated according to empirical relationships and the Stokes law following Tkalich and Chan (2002), dependent on ocean stratification based on temperature and salinity from the ocean model, and the viscosities and densities of oil and water.

In addition to the wave induced entrainment, the oil elements are also subject to vertical turbulence throughout the water column, described using a random-walk scheme based on the turbulent eddy diffusivity from the ocean model.

**Weathering**

In order to calculate weathering of the oil, OpenOil interfaces with the open source ADIOS oil library, developed by NOAA (github.com/NOAA-ORR-ERD/OilLibrary) and (Lehr et al., 2002). In addition to state-of-the-art parametrization of weathering processes such as evaporation, emulsification and dispersion, this software contains a database of measured properties of almost 1000 oil types from around the world. As oils from different sources or wells have vastly different properties, such a database is of vital importance for accurate results. The ADIOS oil library is also used by the NOAA oil drift model github.com/NOAA-ORR-ERD/PyGnome.

The weathering algorithms describes evaporation and emulsification rate of oil, i.e. the water content. The emulsification and evaporation greatly affect oil density, viscosity and oil-water interfacial tension, and thereby the droplet size distribution through Eqs. 1-3. OpenOil take into consideration weathering processes that are dominating in the initial oil spill period of 2-3 days. Long-term weathering processes such as sedimentation and microbial degradation are not considered in this study.

**3   Results**

A first set of simulations are carried out to investigate the effect of oil droplet size distribution. A second set focuses on the effect of river induced fronts. According to Crone and Tolstoy (2010), the average flow rate from the oil well between 22 April and 3 June 2010 was estimated to 0.1 m$^3$ sec$^{-1}$, assuming a liquid oil fraction of 0.4. Gas and highly volatile compounds are not considered here. After the riser was removed and until the leak was sealed on 15 July, the flow rate increased to 0.12 m$^3$ sec$^{-1}$, corresponding to 10,368 m$^3$ day$^{-1}$. An assumption was made on of how much oil was present at the sea surface at the initialization of the simulation. The residence of oil at the sea surface depends heavily on oil properties as well as environmental conditions such as temperature of ocean and air, wind and waves, as described above. According to our mass balance calculations for the DWH spill, using the Light Louisiana Sweet oil type from the NOAA oil library and environmental conditions as described above, it seems reasonable to assume that 80% of the oil mass is removed from the surface after 10 days. This is within the range in our simulations that is typically 60 to 95% (see examples of mass balance plots further down). While Boufadel et al. (2014) assumed a constant removal rate of surface oil 20% per day, the removal rate in the present

simulation and in reality will vary with wind and wave conditions. The simulations for May 2010 are initialized by seeding 48730 particles in a polygon obtained from NOAA shapefiles. Each particle represents initially 1 m$^3$ oil. A continuous point source at the sea floor seeds an additional 8460 particles (8460 m$^{-3}$) per day during the simulation. After June 3rd these numbers are increased by 20% to 10368 m$^3$ day$^{-1}$. The oil elements released at the surface are assigned droplet radii at each entrainment

5    incident, according to the parameterisation of size distributions from respectively DS88 or Li17, see Röhrs et al. (2018) for details. Oil elements at the sea surface (slick) are not considered to have a radius.

Around 20-25 May 2010 there was a significant outflow of the Mississippi River (Fig. 2) and part of the SOP was entrained along the LC resulting in a formation popularly referred to as the "tiger tail" (Fig. 3). The OpenOil simulation shown on top in Fig. 3 is carried out using the classical DS88 oil droplet size distribution (Delvigne and Sweeney, 1988). Fig. 3 lower panel

10   shows the results from repeating this simulation using the new Li17 formulation. In Fig. 4 the mass balance during seven days for the Li17 simulation is shown. There is virtually no difference between DS88 and Li17 and only Li17 is shown here. Both formulations result in about the same fraction of oil at the surface (about 50%) after 7 days, with moderate wind speeds of up to 7.2ms$^{-1}$. The light compounds evaporate fast after release hence the more heavy compounds are tracked here ("dead oil"). Patches of thick oil (where the particles retain nearly 100% of their mass) are visible over a larger area (Fig. 3). Larger oil

15   droplets will rise faster to the surface (North et al., 2011; Röhrs et al., 2018), and DS88 provides a much higher fraction of large droplet after one hour compared to Li17 (Fig. 5, upper panels). However, it turns out that the DS88 and Li17 provide similar volume distributions after a 24 hr test simulation (Fig. 5, lower panels). Still, the peak in the distribution is at around 100 $\mu m$ for both formulations, and rapid rising to the surface can be expected according to North et al. (2011). Due to the small difference between DS88 and Li17, the DS88 simulation results in just marginally more oil stranded after seven days (13.2

20   vs 12.8 %), particularly west of the Mississippi Delta (Figs. 3). A higher fraction of oil at the surface provides more efficient transport by wind and waves towards the shore and larger likelihood of stranding. For both simulations, the oil particles quickly loose 40-50% of their mass, mostly through evaporation.

Fig. 6 shows the geographical distribution of particle diameters at the surface. It is apparent that smaller particles are present outside the MAFLA shelf, probably because the oil particles have been subject to more wind and hence wave action and natural

25   dispersion in the last 24 hours (Fig. 7).

[revised manuscript text omitted]

To the best of our knowledge, this is the first time the importance of the effect of river fronts on oil slick transport in the Gulf of Mexico has been demonstrated using high resolution forcing and a fully fledged oil drift model.

TEXT

*Code availability.* github.com/OpenDrift/opendrift

*Data availability.* TEXT

*Code and data availability.* TEXT

*Sample availability.* TEXT

*Video supplement.* TEXT

**Appendix A**

**A1**

5 *Author contributions.* TEXT

*Competing interests.* TEXT

*Disclaimer.* TEXT

*Acknowledgements.* This research was made possible by a grant from The Gulf of Mexico Research Initiative (award "Influence of river induced fronts on hydrocarbon transport," GOMA 23160700). Atmospheric and wave data were kindly provided by the European Center 10 for Medium-Range Weather Forecasts (ECMWF). M. Le Hénaff acknowledges partial support from the Physical Oceanography Division at NOAA's Atlantic Oceanographic and Meteorological Laboratory, AOML. The outputs from the HYCOM simulations are available at the Gulf of Mexico Research Initiative Information and Data Cooperative (GRIIDC - doi:10.7266/N7NG4NPC).

[revised manuscript text omitted]

---

## Editor Comment (EC1) · Piers Chapman (Editor) · 27 Feb 2019

I thank the authors for making substantial changes to their manuscript as a result of the comments of the reviewers. As these comments were generally very technical, I have asked the reviewers to read the final submitted text and let me know if they are satisfied by the changes.

I apologize for taking longer than hoped to respond, but I was ill for much of the past week.

Piers Chapman